# CONCEPT-BASED EXPLANATIONS FOR OUT-OF-DISTRIBUTION DETECTORS

## ABSTRACT

Out-of-distribution (OOD) detection plays a crucial role in ensuring the safe deployment of deep neural network (DNN) classifiers. While a myriad of methods have focused on improving the performance of OOD detectors, a critical gap remains in interpreting their decisions. We help bridge this gap by providing explanations for OOD detectors based on learned high-level concepts. We first propose two new metrics for assessing the effectiveness of a particular set of concepts for explaining OOD detectors: 1) *detection completeness*, which quantifies the sufficiency of concepts for explaining an OOD-detector's decisions, and 2) *concept separability*, which captures the distributional separation between in-distribution and OOD data in the concept space. Based on these metrics, we propose a framework for learning a set of concepts that satisfy the desired properties of detection completeness and concept separability, and demonstrate the framework's effectiveness in providing concept-based explanations for diverse OOD detection techniques. We also show how to identify prominent concepts that contribute to the detection results via a modified Shapley value-based importance score.

## 1 INTRODUCTION

It is well known that machine learning (ML) models can yield uncertain and unreliable predictions on out-of-distribution (OOD) inputs, *i.e.*, inputs from outside the training distribution (Amodei et al., 2016; Goodfellow et al., 2015; Nguyen & O'Connor, 2015). The most common line of defense in this situation is to augment the ML model (*e.g.*, a DNN classifier) with a detector that can identify and flag such inputs as OOD. The ML model can then abstain from making predictions on such inputs (Hendrycks et al., 2019; Lin et al., 2021; Mohseni et al., 2020).

Currently, the problem of explaining the decisions of an OOD detector remains largely unexplored. Much of the focus in learning OOD detectors has been on improving their detection performance (Liu et al., 2020; Mohseni et al., 2020; Lin et al., 2021; Chen et al., 2021; Sun et al., 2021; Cao & Zhang, 2022), but not on improving their explainability. A potential approach would be to run an existing interpretation method for DNN classifiers with ID and OOD data separately, and then inspect the difference between the generated explanations. However, it is not known if an explanation method that is effective for explaining in-distribution class predictions will also be effective for OOD detectors. For instance, feature attributions, the most popular type of explanation (Sundararajan et al., 2017; Ribeiro et al., 2016), may not capture visual differences in the generated explanations between ID and OOD inputs (Adebayo et al., 2020). Moreover, their explanations based on pixel-level activations may not provide the most intuitive form of explanations for humans.

This paper addresses the above research gap by proposing the first method (to our knowledge) to help interpret the decisions of an OOD detector in a human-understandable way. We build upon recent advances in concept-based explanations for DNN classifiers (Ghorbani et al., 2019; Zhou et al., 2018a; Bouchacourt & Denoyer, 2019; Yeh et al., 2020), which offer an advantage of providing explanations in terms of high-level *concepts* for classification tasks. We make the first effort at extending their utility to the problem of OOD detection. Consider Figure 1 which illustrates our concept-based explanations given inputs which are all classified as "Dolphin" by a DNN classifier, but detected as either ID or OOD by an OOD detector. We observe that the OOD detector predicts a certain input as ID when its concept-score patterns are similar to that of ID images from the Dolphin class. Likewise, the detector predicts an input as OOD when its concept-score patterns are very different from that of

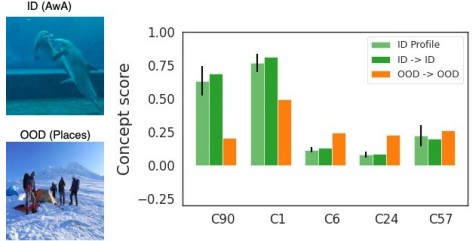

(a) Correct detection: ID (or OOD) dolphin image correctly detected as ID (or OOD).

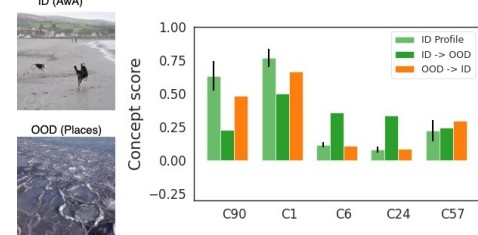

(b) Wrong detection: ID (or OOD) dolphin image falsely detected as OOD (or ID).

Figure 1: **Our concept-based explanation for the Energy OOD detector (Liu et al., 2020).** On the x-axis, we present the top-5 important concepts that describe the detector's behavior given images classified as "Dolphin". Concepts C90 and C1 represent "dolphin-like skin" and "wavy surface of the sea" respectively (see Figure 12b). ID profile shows the concept score patterns for ID images predicted as Dolphin.

ID inputs. A user can verify whether the OOD detector makes decisions based on the concepts that are aligned with human intuition (*e.g.*, C90 and C1), and that the incorrect detection (as in Figure 1b) is an understandable mistake, not a misbehavior of the OOD detector. Such explanations can help the user evaluate the reliability of the OOD detector and decide upon its adoption in practice.

We aim to provide a general interpretability framework that is applicable to the wide range of OOD detectors in the world. Accordingly, a research question we ask is: *without relying on the internal mechanism of an OOD detector, can we determine a good set of concepts that are appropriate for understanding why the OOD detector predicts a certain input to be ID/OOD?* A key contribution of this paper is to show that this can be done in an unsupervised manner without any human annotations.

We make the following contributions in this paper:

- We motivate and propose new metrics to quantify the effectiveness of concept-based explanations for a black-box OOD detector, namely *detection completeness* and *concept separability* (§ 2.2, § 3.1, and § 3.2).
- We propose a concept-learning objective with suitable regularization terms that, given an OOD detector for a DNN classifier, learns a set of concepts with good detection completeness and concept separability (§ 3.3);
- By treating a given OOD detector as black-box, we show that our method can be applied to explain a variety of existing OOD detection methods. By identifying prominent concepts that contribute to an OOD detector's decisions via a modified Shapley value score based on the detection completeness, we demonstrate how the discovered concepts can be used to understand the OOD detector. (§ 4).

**Related Work.** In the literature of OOD detection, recent studies have designed various scoring functions based on the representation from the final or penultimate layers (Liang et al., 2018; DeVries & Taylor, 2018), or a combination of different internal layers of DNN model (Lee et al., 2018; Lin et al., 2021; Raghuram et al., 2021). A recent survey on generalized OOD detection can be found in Yang et al. (2021). Our work aims to provide post-hoc explanations applicable to a wide range of black-box OOD detectors without modifying their internals. Among different interpretability approaches, concept-based explanation (Koh et al., 2020; Alvarez Melis & Jaakkola, 2018) has gained popularity as it is designed to be better-aligned with human reasoning (Armstrong et al., 1983; Tenenbaum, 1999) and intuition (Ghorbani et al., 2019; Zhou et al., 2018a; Bouchacourt & Denoyer, 2019; Yeh et al., 2020). There have been limited attempts to assess the use of concept-based explanations under data distribution changes such as adversarial manipulation (Kim et al., 2018) or spurious correlations (Adebayo et al., 2020). Designing concept-based explanations for OOD detection requires further exploration and is the focus of our work.

## 2 PROBLEM SETUP AND BACKGROUND

**Notations.** Let $\mathcal{X} \subseteq \mathbb{R}^{a_0 \times b_0 \times d_0}$ denote the space of inputs [1] $\mathbf{x}$, where $d_0$ is the number of channels and $a_0$ and $b_0$ are the image size along each channel. Let $\mathcal{Y} := \{1, \cdots, L\}$ denote the space of

---

[1] We focus on images, but the proposed method extends to other domains.

output class labels $y$. Let $\Delta_L$ denote the set of all probabilities over $\mathcal{Y}$ (the simplex in $L$-dimensions). We assume that natural inputs to the DNN classifier are sampled from an unknown probability distribution $P_{\text{in}}$ over the space $\mathcal{X} \times \mathcal{Y}$. The compact notation $[n]$ denotes $\{1, \cdots, n\}$ for a positive integer $n$. Boldface symbols are used to denote both vectors and tensors. $\langle \mathbf{x}, \mathbf{x}' \rangle$ denotes the standard inner-product between a pair of vectors. The indicator function $\mathbb{1}[c]$ takes value 1 (0) when the condition $c$ is true (false).

**ID and OOD Datasets.** Consider a labeled ID training dataset $D_{\text{in}}^{\text{tr}} = \{(\mathbf{x}_i, y_i), \ i = 1, \cdots, N_{\text{in}}^{\text{tr}}\}$ sampled from the distribution $P_{\text{in}}$. We assume the availability of an unlabeled training dataset $D_{\text{out}}^{\text{tr}} = \{\widetilde{\mathbf{x}}_i, \ i = 1, \cdots, N_{\text{out}}^{\text{tr}}\}$ from a different distribution, referred to as the *auxiliary OOD dataset*. Similarly, we define the ID test dataset (sampled from $P_{\text{in}}$) as $D_{\text{in}}^{\text{te}}$, and the OOD test dataset as $D_{\text{out}}^{\text{te}}$. Note that the auxiliary OOD dataset $D_{\text{in}}^{\text{tr}}$ and the test OOD dataset $D_{\text{out}}^{\text{te}}$ are from different distributions. All the OOD datasets are unlabeled since their label space is usually different from $\mathcal{Y}$.

**OOD Detector.** The goal of an OOD detector is to determine if a test input to the classifier is ID (*i.e.*, from the distribution $P_{\text{in}}$); otherwise the input is declared to be OOD (Yang et al., 2021). Given a trained classifier $\mathbf{f} : \mathcal{X} \mapsto \Delta_L$, the decision function of an OOD detector can be generally defined as $\mathcal{D}_\gamma(\mathbf{x}, \mathbf{f}) = \mathbb{1}[S(\mathbf{x}, \mathbf{f}) \geq \gamma]$, where $S(\mathbf{x}, \mathbf{f}) \in \mathbb{R}$ is the score function of the detector for an input $\mathbf{x}$ and $\gamma$ is the threshold. We follow the convention that larger scores correspond to ID inputs, and the detector outputs of 1 and 0 correspond to ID and OOD respectively. We assume the availability a pre-trained DNN classifier and a paired OOD detector that is trained to detect inputs for the classifier.

## 2.1 PROJECTION INTO CONCEPT SPACE

Consider a pre-trained DNN classifier $\mathbf{f} : \mathcal{X} \mapsto \Delta_L$ that maps an input $\mathbf{x}$ to its corresponding predicted class probabilities. Without loss of generality, we can partition the DNN at a convolutional layer $\ell$ into two parts, *i.e.*, $\mathbf{f} = \mathbf{h} \circ \phi$ where: 1) $\phi : \mathcal{X} \mapsto \mathcal{Z} := \mathbb{R}^{a_\ell b_\ell \times d_\ell}$ is the first half of $\mathbf{f}$ that maps an input $\mathbf{x}$ to the intermediate feature representation [2] $\phi(\mathbf{x})$, and 2) $\mathbf{h} : \mathcal{Z} \mapsto \Delta_L$ is the second half of $\mathbf{f}$ that maps $\phi(\mathbf{x})$ to the predicted class probabilities $\mathbf{h}(\phi(\mathbf{x}))$. We denote the predicted probability of a class $y$ by $f_y(\mathbf{x}) = h_y(\phi(\mathbf{x}))$, and the prediction of the classifier by $\widehat{y}(\mathbf{x}) = \text{argmax}_y f_y(\mathbf{x})$.

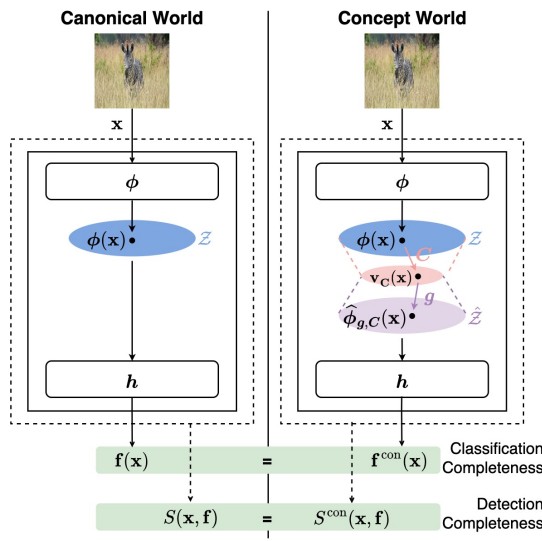

Figure 2: Our two-world view of the classifier and OOD detector.

Our work is based on the common implicit assumption of linear interpretability in the concept-based explanation literature, *i.e.*, high-level concepts lie in a linearly-projected subspace of the feature representation space $\mathcal{Z}$ of the classifier (Kim et al., 2018). Consider a projection matrix $\mathbf{C} = [\mathbf{c}_1, \cdots, \mathbf{c}_m] \in \mathbb{R}^{d_\ell \times m}$ (with $m \ll d_\ell$) that maps from the space $\mathcal{Z}$ into a reduced-dimension concept space. $\mathbf{C}$ consists of $m$ unit vectors, where $\mathbf{c}_i \in \mathbb{R}^{d_\ell}$ is referred to as the *concept vector* representing the $i$-th concept (*e.g.*, "stripe" or "oval face"), and $m$ is the number of concepts. We define the *concept score* for $\mathbf{x}$ as the linear projection of the high-dimensional layer representation $\phi(\mathbf{x}) \in \mathbb{R}^{a_\ell b_\ell \times d_\ell}$ into the concept space (Yeh et al., 2020), i.e. $\mathbf{v_C}(\mathbf{x}) := \phi(\mathbf{x}) \mathbf{C} \in \mathbb{R}^{a_\ell b_\ell \times m}$. We also define a mapping from the projected concept space back to the feature space by a non-linear function $\mathbf{g} : \mathbb{R}^{a_\ell b_\ell \times m} \mapsto \mathbb{R}^{a_\ell b_\ell \times d_\ell}$. The reconstructed feature representation at layer $\ell$ is then defined as $\widehat{\phi}_{\mathbf{g},\mathbf{C}}(\mathbf{x}) := \mathbf{g}(\mathbf{v_C}(\mathbf{x}))$.

---

[2]We flatten the first two dimensions of the feature representation, thus changing an $a_\ell \times b_\ell \times d_\ell$ tensor to an $a_\ell b_\ell \times d_\ell$ matrix, where $a_\ell$ and $b_\ell$ are the filter size and $d_\ell$ is the number of channels.

## 2.2 CANONICAL WORLD AND CONCEPT WORLD

As shown in Fig. 2, we consider a "two-world" view of the classifier and OOD detector consisting of the *canonical world* and the *concept world*, which are defined as follows:

**Canonical World.** In this case, both the classifier and OOD detector use the original layer representation $\phi(\mathbf{x})$ for their predictions. The prediction of the classifier is $\mathbf{f}(\mathbf{x}) = \mathbf{h}(\phi(\mathbf{x}))$, and the decision function of the detector is $\mathcal{D}_\gamma(\mathbf{x}, \mathbf{h} \circ \phi)$ with a score function $S(\mathbf{x}, \mathbf{h} \circ \phi)$.

**Concept World.** We use the following observation in constructing the concept-world formulation: *both the classifier and the OOD detector can be modified to make predictions based on the reconstructed feature representation*, i.e., using $\widehat{\phi}_{\mathbf{g}, \mathbf{C}}(\mathbf{x})$ instead of $\phi(\mathbf{x})$. Accordingly, we define the corresponding classifier, detector, and score function in the concept world as follows:

$$
\begin{aligned}
\mathbf{f}^{\mathrm{con}}(\mathbf{x}) &:= \mathbf{h}(\widehat{\phi}_{\mathbf{g}, \mathbf{C}}(\mathbf{x})) = \mathbf{h}(\mathbf{g}(\mathbf{v}_{\mathbf{C}}(\mathbf{x}))) \\
\mathcal{D}_\gamma^{\mathrm{con}}(\mathbf{x}, \mathbf{f}) &:= \mathcal{D}_\gamma(\mathbf{x}, \mathbf{h} \circ \widehat{\phi}_{\mathbf{g}, \mathbf{C}}) = \mathcal{D}_\gamma(\mathbf{x}, \mathbf{h} \circ \mathbf{g} \circ \mathbf{v}_{\mathbf{C}}) \\
S^{\mathrm{con}}(\mathbf{x}, \mathbf{f}) &:= S(\mathbf{x}, \mathbf{h} \circ \widehat{\phi}_{\mathbf{g}, \mathbf{C}}) = S(\mathbf{x}, \mathbf{h} \circ \mathbf{g} \circ \mathbf{v}_{\mathbf{C}}).
\end{aligned}
\tag{1}
$$

We further elaborate on this two-world view and introduce the following two desirable properties.

**Detection Completeness.** Given a fixed algorithmic approach for learning the classifier and OOD detector, and with fixed internal parameters of $\mathbf{f}$, we would ideally like the classifier prediction and the detection score to be indistinguishable between the two worlds. In other words, for the concepts to *sufficiently* explain the OOD detector, we require $\mathcal{D}_\gamma^{\mathrm{con}}(\mathbf{x}, \mathbf{f})$ to closely mimic $\mathcal{D}_\gamma(\mathbf{x}, \mathbf{f})$. Likewise, we require $\mathbf{f}^{\mathrm{con}}(\mathbf{x})$ to closely mimic $\mathbf{f}(\mathbf{x})$ since the detection mechanism of $\mathcal{D}_\gamma$ is closely paired to the classifier. We refer to this property as the *completeness of a set of concepts with respect to the OOD detector and its paired classifier*. As discussed in § 3.1, this extends the notion of classification completeness introduced by Yeh et al. (2020) to an OOD detector and its paired classifier.

**Concept Separability.** To improve the interpretability of the resulting explanations for the OOD detector, we require another desirable property from the learned concepts: data detected as ID by $\mathcal{D}_\gamma$ (henceforth referred to as *detected-ID* data) and data detected as OOD by $\mathcal{D}_\gamma$ (henceforth referred to as *detected-OOD* data) should be well-separated in the concept-score space. Since our goal is to help an analyst understand which concepts distinguish the detected-ID data from detected-OOD data, we would like to learn a set of concepts that have a well-separated concept score pattern for inputs from these two groups (*e.g.*, the concepts "stripe" and "oval face" in Fig. **??** have distinct concept scores).

## 3 PROPOSED APPROACH

Given a trained DNN classifier $\mathbf{f}$, a paired OOD detector $\mathcal{D}_\gamma$, and a set of concepts $\mathbf{C}$, we address the following questions: **1)** *Are the concepts sufficient to capture the prediction behavior of both the classifier and OOD detector?* (see § 3.1); **2)** *Do the concepts show clear distinctions in their scores between detected-ID data and detected-OOD data?* (see § 3.2). We first propose new metrics for quantifying the set of learned concepts, followed by a general framework for learning concepts that possess these properties (see § 3.3).

### 3.1 METRICS FOR DETECTION COMPLETENESS

**Definition 1.** Given a trained DNN classifier $\mathbf{f} = \mathbf{h} \circ \phi$ and a set of concept vectors $\mathbf{C}$, the *classification completeness* with respect to $P_{\mathrm{in}}(\mathbf{x}, y)$ is defined as (Yeh et al., 2020):

$$
\eta_{\mathbf{f}}(\mathbf{C}) := \frac{\sup_{\mathbf{g}} \mathbb{E}_{(\mathbf{x}, y) \sim P_{\mathrm{in}}} \left[ \mathbb{1}[y = \arg\max_{y'} h_{y'}(\widehat{\phi}_{\mathbf{g}, \mathbf{C}}(\mathbf{x}))] \right] - a_r}{\mathbb{E}_{(\mathbf{x}, y) \sim P_{\mathrm{in}}} \left[ \mathbb{1}[y = \arg\max_{y'} h_{y'}(\phi(\mathbf{x}))] \right] - a_r}
$$

where $a_r = 1/L$ is the accuracy of a random $L$-class classifier.

The denominator of $\eta_{\mathbf{f}}(\mathbf{C})$ is the accuracy of the original classifier $\mathbf{f}$, while the numerator is the maximum accuracy that can be achieved in the concept world using the feature representation reconstructed from the concept scores. The maximization is over the parameters of the neural network $\mathbf{g}$ that reconstructs the feature representation from the vector of concept scores.

**Definition 2.** Given a trained DNN classifier $\mathbf{f} = \mathbf{h} \circ \phi$, a trained OOD detector with score function $S(\mathbf{x}, \mathbf{f})$, and a set of concept vectors $\mathbf{C}$, we define the *detection completeness score* with respect to the ID distribution $P_{\text{in}}(\mathbf{x}, y)$ and OOD distribution $P_{\text{out}}(\mathbf{x})$ as follows:

$$\eta_{\mathbf{f}, S}(\mathbf{C}) := \frac{\sup_{\mathbf{g}} \text{AUC}(\mathbf{h} \circ \widehat{\phi}_{\mathbf{g}, \mathbf{C}}) - b_r}{\text{AUC}(\mathbf{h} \circ \phi) - b_r}, \tag{2}$$

where $\text{AUC}(\mathbf{f})$ is the area under the ROC curve of an OOD detector based on $\mathbf{f}$, defined as $\text{AUC}(\mathbf{f}) := \mathbb{E}_{(\mathbf{x}, y) \sim P_{\text{in}}} \mathbb{E}_{\mathbf{x}' \sim P_{\text{out}}} \mathbb{1}\big[S(\mathbf{x}, \mathbf{f}) > S(\mathbf{x}', \mathbf{f})\big]$, and $b_r = 0.5$ is the AUROC of a random detector.

The numerator term is the maximum achievable AUROC in the concept world via reconstructed features from concept scores. In practice, $\text{AUC}(\mathbf{f})$ is estimated using the test datasets $D_{\text{in}}^{\text{te}}$ and $D_{\text{out}}^{\text{te}}$. Both the classification completeness and detection completeness scores are designed to be in the range $[0, 1]$. However, this is not strictly guaranteed since the classifier or OOD detector in the concept world may empirically have a better (corresponding) metric on a given ID/OOD dataset. A completeness score close to 1 indicates that the set of concepts $\mathbf{C}$ are close to complete in characterizing the behavior of the classifier and/or the OOD detector.

## 3.2 CONCEPT SEPARABILITY SCORE

**Concept Scores.** In Section 2.1, we introduced a projection matrix $\mathbf{C} \in \mathbb{R}^{d_\ell \times m}$ that maps $\phi(\mathbf{x})$ to $\mathbf{v}_{\mathbf{C}}(\mathbf{x})$, and consists of $m$ unit concept vectors $\mathbf{C} = [\mathbf{c}_1 \cdots \mathbf{c}_m]$. The inner product between the feature representation and a concept vector is referred to as the *concept score*, and it quantifies how close an input is to the given concept (Kim et al., 2018; Ghorbani et al., 2019). Specifically, the concept score corresponding to concept $i$ is defined as $\mathbf{v}_{\mathbf{c}_i}(\mathbf{x}) := \langle \phi(\mathbf{x}), \mathbf{c}_i \rangle = \phi(\mathbf{x}) \mathbf{c}_i \in \mathbb{R}^{a_\ell b_\ell}$. The matrix of concept scores from all the concepts is simply the concatenation of the individual concept scores, *i.e.*, $\mathbf{v}_{\mathbf{C}}(\mathbf{x}) = \phi(\mathbf{x}) \mathbf{C} = [\mathbf{v}_{\mathbf{c}_1}(\mathbf{x}) \cdots \mathbf{v}_{\mathbf{c}_m}(\mathbf{x})] \in \mathbb{R}^{a_\ell b_\ell \times m}$. We also define a dimension-reduced version of the concept scores that takes the maximum of the inner-product over each $a_\ell \times b_\ell$ patch as follows: $\widetilde{\mathbf{v}}_{\mathbf{C}}(\mathbf{x})^T = [\widetilde{v}_{\mathbf{c}_1}(\mathbf{x}), \cdots, \widetilde{v}_{\mathbf{c}_m}(\mathbf{x})] \in \mathbb{R}^m$, where $\widetilde{v}_{\mathbf{c}_i}(\mathbf{x}) = \max_{p,q} |\langle \phi^{p,q}(\mathbf{x}), \mathbf{c}_i \rangle| \in \mathbb{R}$. Here $\phi^{p,q}(\mathbf{x})$ is the feature representation corresponding to the $(p, q)$-th patch of input $\mathbf{x}$ (*i.e.*, receptive field (Araujo et al., 2019)). This reduction operation is done to capture the most important correlations from each patch, and the $m$-dimensional concept score will be used to define our concept separability metric as follows.

We would like the set of concept-score vectors from the detected-ID class $V_{\text{in}}(\mathbf{C}) := \{\widetilde{\mathbf{v}}_{\mathbf{C}}(\mathbf{x}), \mathbf{x} \in D_{\text{in}}^{\text{tr}} \cup D_{\text{out}}^{\text{tr}} : \mathcal{D}_\gamma(\mathbf{x}, \mathbf{f}) = 1\}$, and the set of concept-score vectors from the detected-OOD class $V_{\text{out}}(\mathbf{C}) := \{\widetilde{\mathbf{v}}_{\mathbf{C}}(\mathbf{x}), \mathbf{x} \in D_{\text{in}}^{\text{tr}} \cup D_{\text{out}}^{\text{tr}} : \mathcal{D}_\gamma(\mathbf{x}, \mathbf{f}) = 0\}$ to be well separated. Let $J_{\text{sep}}(V_{\text{in}}(\mathbf{C}), V_{\text{out}}(\mathbf{C})) \in \mathbb{R}$ define a general *measure of separability* between the two subsets, such that a larger value corresponds to higher separability. We discuss a specific choice for $J_{\text{sep}}$ for which it is possible to tractably optimize concept separability as part of the learning objective in Section 3.3.

**Global Concept Separability.** Class separability metrics have been well studied in the pattern recognition literature, particularly for the two-class case (Fukunaga, 1990b)[3]. Motivated by Fisher's linear discriminant analysis (LDA), we explore the use of class-separability measures based on the within-class and between-class scatter matrices (Murphy, 2012). The goal of LDA is to find a projection vector (direction) such that data from the two classes are maximally separated and form compact clusters upon projection. Rather than finding an optimal projection direction, we are more interested in ensuring that the concept-score vectors from the detected-ID and detected-OOD data have high separability. Consider the within-class and between-class scatter matrices based on $V_{\text{in}}(\mathbf{C})$ and $V_{\text{out}}(\mathbf{C})$, given by

$$\mathbf{S}_w = \sum_{\mathbf{v} \in V_{\text{in}}(\mathbf{C})} (\mathbf{v} - \boldsymbol{\mu}_{\text{in}})(\mathbf{v} - \boldsymbol{\mu}_{\text{in}})^T + \sum_{\mathbf{v} \in V_{\text{out}}(\mathbf{C})} (\mathbf{v} - \boldsymbol{\mu}_{\text{out}})(\mathbf{v} - \boldsymbol{\mu}_{\text{out}})^T, \tag{3}$$

$$\mathbf{S}_b = (\boldsymbol{\mu}_{\text{out}} - \boldsymbol{\mu}_{\text{in}})(\boldsymbol{\mu}_{\text{out}} - \boldsymbol{\mu}_{\text{in}})^T, \tag{4}$$

where $\boldsymbol{\mu}_{\text{in}}$ and $\boldsymbol{\mu}_{\text{out}}$ are the mean concept-score vectors from $V_{\text{in}}(\mathbf{C})$ and $V_{\text{out}}(\mathbf{C})$ respectively. We define the following separability metric based on the generalized eigenvalue equation solved by Fisher's LDA (Fukunaga, 1990b): $J_{\text{sep}}(\mathbf{C}) := J_{\text{sep}}(V_{\text{in}}(\mathbf{C}), V_{\text{out}}(\mathbf{C})) = \text{tr}\big[\mathbf{S}_w^{-1} \mathbf{S}_b\big]$. Maximizing the above metric is equivalent to maximizing the sum of eigenvalues of the matrix $\mathbf{S}_w^{-1} \mathbf{S}_b$, which

---

[3]In our problem, the two classes correspond to detected-ID and detected-OOD.

in-turn ensures a large between-class separability and a small within-class separability for the detected-ID and detected-OOD concept scores. We refer to this as a *global concept separability* metric because it does not analyze the separability on a per-class level [4]. The separability metric is closely related to the Bhattacharya distance, which is an upper bound on the Bayes error rate (see Appendix A.1).

### 3.3 PROPOSED CONCEPT LEARNING – KEY IDEAS

**Prior Approaches and Limitations.** Among post-hoc concept-discovery methods for a DNN classifier with ID data, unlike Kim et al. and Ghorbani et al., that do not support imposing required conditions into the concept discovery, Yeh et al. devised a learning-based approach where classification completeness and the saliency of concepts are optimized via a regularized objective given by

$$\underset{\mathbf{C},\mathbf{g}}{\operatorname{argmax}} \ \underset{(\mathbf{x},y)\sim P_{\text{in}}}{\mathbb{E}} \big[ \log h_y(\mathbf{g}(\mathbf{v_C}(\mathbf{x}))) \big] \ + \ \lambda_{\text{expl}} \, R_{\text{expl}}(\mathbf{C}). \tag{5}$$

Here $\mathbf{C}$ and $\mathbf{g}$ (parameterized by a neural network) are jointly optimized, and $R_{\text{expl}}(\mathbf{C})$ is a regularization term used to ensure that the learned concept vectors have high spatial coherency and low redundancy among themselves (see (Yeh et al., 2020) for the definition).

While the objective (5) of Yeh et al. can learn a set of sufficient concepts that have a high classification completeness score, we find that it does not necessarily replicate the per-instance prediction behavior of the classifier in the concept world. Specifically, there can be discrepancies in the reconstructed feature representation, whose effect propagates through the remaining part of the classifier. Since many widely-used OOD detectors rely on the feature representations and/or the classifier's predictions, this discrepancy in the existing concept learning approaches makes it hard to closely replicate the OOD detector in the concept world (see Fig. 3). Furthermore, the scope of Yeh et al. is confined to concept learning for explaining the classifier's predictions based on ID data, and there is no guarantee that the learned concepts would be useful for explaining an OOD detector. To address these gaps, we propose a general method for concept learning that complements prior work by imposing additional instance-level constraints on the concepts, and by considering both the OOD detector and OOD data.

**Concept Learning Objective.** We define a concept learning objective that aims to find a set of concepts $\mathbf{C}$ and a mapping $\mathbf{g}$ that have the following properties: 1) high detection completeness w.r.t the OOD detector; 2) high classification completeness w.r.t the DNN classifier; and 3) high separability in the concept-score space between detected-ID data and detected-OOD data.

Inspired by recent works on transferring feature information from a teacher model to a student model (Hinton et al., 2015; Zhou et al., 2018b), we encourage accurate reconstruction of $\hat{\mathcal{Z}}$ based on the concept scores by adding a regularization term that is the squared $\ell_2$ distance between the original and reconstructed representations $J_{\text{norm}}(\mathbf{C},\mathbf{g}) = \mathbb{E}_{\mathbf{x}\sim P_{\text{in}}} \|\phi(\mathbf{x}) - \widehat{\phi}_{\mathbf{g},\mathbf{C}}(\mathbf{x})\|^2$. In order to close the gap between the scores of the OOD detector in the concept world and canonical world on a per-sample level, we introduce the following mean-squared-error (MSE) based regularization:

$$J_{\text{mse}}(\mathbf{C},\mathbf{g}) = \underset{\mathbf{x}\sim P_{\text{in}}}{\mathbb{E}} \big(S(\mathbf{x},\mathbf{h}\circ\widehat{\phi}_{\mathbf{g},\mathbf{C}}) - S(\mathbf{x},\mathbf{f})\big)^2 + \underset{\mathbf{x}\sim P_{\text{out}}}{\mathbb{E}} \big(S(\mathbf{x},\mathbf{h}\circ\widehat{\phi}_{\mathbf{g},\mathbf{C}}) - S(\mathbf{x},\mathbf{f})\big)^2. \tag{6}$$

MSE terms are computed with both the ID and OOD data because we want to ensure that the ROC curve corresponding to both the score functions are close to each other (which requires OOD data). Finally, we include a regularization term to maximize the separability metric between the detected-ID and detected-OOD data in the concept-score space, resulting in our final concept learning objective:

$$\underset{\mathbf{C},\mathbf{g}}{\operatorname{argmax}} \ \underset{(\mathbf{x},y)\sim P_{\text{in}}}{\mathbb{E}} \big[ \log h_y(\mathbf{g}(\mathbf{v_C}(\mathbf{x}))) \big] \ + \ \lambda_{\text{expl}} \, R_{\text{expl}}(\mathbf{C})$$
$$- \ \lambda_{\text{mse}} \, J_{\text{mse}}(\mathbf{C},\mathbf{g}) \ - \ \lambda_{\text{norm}} \, J_{\text{norm}}(\mathbf{C},\mathbf{g}) \ + \ \lambda_{\text{sep}} \, J_{\text{sep}}(\mathbf{C}). \tag{7}$$

The $\lambda$ coefficients are non-negative hyper-parameters that are further discussed in Section 4. We note that both $J_{\text{mse}}(\mathbf{C},\mathbf{g})$ and $J_{\text{sep}}(\mathbf{C})$ depend on the OOD detector [5]. We use the SGD-based Adam optimizer (Kingma & Ba, 2014)) to solve the learning objective. The expectations involved in the objective terms are calculated using sample estimates from the training ID and OOD datasets. Specifically, $D_{\text{in}}^{\text{tr}}$ and $D_{\text{out}}^{\text{tr}}$ are used to compute the expectations over $P_{\text{in}}$ and $P_{\text{out}}$, respectively. Our complete concept learning is summarized in Algorithm 1 (Appendix A.4).

---

[4]See Appendix A.2 and A.3 for per-class variations of detection completeness and concept separability.

[5]This dependence may not be obvious for the separability term, but it is clear from its definition.

## 4 EXPERIMENTS

We briefly describe the experimental setup here and provide additional details in Appendix B.

**Datasets.** For the ID dataset, we use Animals with Attributes (AwA) (Xian et al., 2018) with 50 animal classes, and split it into a train set (29841 images), validation set (3709 images), and test set (3772 images). We use the MSCOCO dataset (Lin et al., 2014) as the auxiliary OOD dataset $D_{\text{out}}^{\text{tr}}$ for training and validation. For the OOD test dataset $D_{\text{out}}^{\text{te}}$, we follow a common setting in the literature of large-scale OOD detection (Huang & Li, 2021) and use three different image datasets: Places365 (Zhou et al., 2017), SUN (Xiao et al., 2010), and Textures (Cimpoi et al., 2014).

**Models.** We apply our framework to interpret five prominent OOD detectors from the literature: MSP (Hendrycks & Gimpel, 2016), ODIN (Liang et al., 2018), Generalized-ODIN (Hsu et al., 2020), Energy (Liu et al., 2020) and Mahalanobis (Lee et al., 2018). The OOD detectors are paired with the widely-used Inception-V3 model (Szegedy et al., 2016) (following the setup in prior works (Yeh et al., 2020; Ghorbani et al., 2019; Kim et al., 2018)) trained on the Animals-with-Attributes (AwA) dataset (Xian et al., 2018), which has a test accuracy of 92.10%.

**Metrics.** For each set of concepts learned with different OOD detectors and hyperparameters, we report the classification completeness $\eta_{\mathbf{f}}(\mathbf{C})$, detection completeness $\eta_{\mathbf{f},S}(\mathbf{C})$, and the relative concept separability metric (defined below). In contrast to the completeness scores that are almost always bounded to the range $[0, 1]$, it is hard to gauge the possible range of the separability score $J_{\text{sep}}(\mathbf{C})$ (or $J_{\text{sep}}^{y}(\mathbf{C})$) across different settings (datasets, classification models, and OOD detectors), and whether the value represents a significant improvement in separability. Hence, we define the *relative concept separability*, which captures the relative improvement in concept separability using concepts $\mathbf{C}$ compared to a different set of concepts $\mathbf{C}'$, as follows

$$J_{\text{sep}}(\mathbf{C}, \mathbf{C}') = \text{Median}\left(\left\{\frac{J_{\text{sep}}^{y}(\mathbf{C}) - J_{\text{sep}}^{y}(\mathbf{C}')}{J_{\text{sep}}^{y}(\mathbf{C}')}\right\}_{y=1}^{L}\right). \tag{8}$$

We choose $\mathbf{C}'$ to be the set of concepts learned by the baseline (Yeh et al., 2020), which is a special case of our learning objective when $\lambda_{\text{mse}} = \lambda_{\text{norm}} = \lambda_{\text{sep}} = 0$. The set of concept $\mathbf{C}$ are obtained via our concept learning objective, with various combinations of hyperparameter values.

| OOD detector | Hyper-parameters | $\eta_{\mathbf{f}}(\mathbf{C})$ ↑ | Places | | SUN | | Textures | |
|---|---|---|---|---|---|---|---|---|
| | | | $\eta_{\mathbf{f},S}(\mathbf{C})$ ↑ | $J_{\text{sep}}(\mathbf{C},\mathbf{C}')$ ↑ | $\eta_{\mathbf{f},S}(\mathbf{C})$ ↑ | $J_{\text{sep}}(\mathbf{C},\mathbf{C}')$ ↑ | $\eta_{\mathbf{f},S}(\mathbf{C})$ ↑ | $J_{\text{sep}}(\mathbf{C},\mathbf{C}')$ ↑ |
| MSP | $(0,0,0)$ | $0.977 \pm 0.0006$ | $0.774 \pm 0.0010$ | $0.694 \pm 0.0153$ | $0.782 \pm 0.0010$ | $1.088 \pm 0.0175$ | $0.593 \pm 0.0013$ | $0.765 \pm 0.0157$ |
| | $(10,0.1,0)$ | $\mathbf{0.994} \pm 0.0004$ | $\underline{0.947} \pm 0.0004$ | $1.892 \pm 0.0393$ | $\underline{0.946} \pm 0.0004$ | $3.074 \pm 0.0531$ | $\underline{0.920} \pm 0.0005$ | $\underline{3.577} \pm 0.1292$ |
| | $(0,0,50)$ | $0.980 \pm 0.0005$ | $0.814 \pm 0.0008$ | $\underline{2.533} \pm 0.0714$ | $0.816 \pm 0.0009$ | $\underline{4.295} \pm 0.1048$ | $0.773 \pm 0.0010$ | $3.147 \pm 0.2076$ |
| | $(10,0.1,50)$ | $\underline{0.984} \pm 0.0004$ | $\mathbf{0.960} \pm 0.0004$ | $\mathbf{2.756} \pm 0.0854$ | $\mathbf{0.961} \pm 0.0005$ | $\mathbf{4.442} \pm 0.0830$ | $\mathbf{0.937} \pm 0.0004$ | $\mathbf{3.587} \pm 0.2145$ |
| ODIN | $(0,0,0)$ | $0.977 \pm 0.0006$ | $0.742 \pm 0.0011$ | $0.444 \pm 0.0119$ | $0.745 \pm 0.0010$ | $0.710 \pm 0.0156$ | $0.618 \pm 0.0013$ | $0.501 \pm 0.0121$ |
| | $(10^8,0.1,0)$ | $\mathbf{0.994} \pm 0.0004$ | $\underline{0.951} \pm 0.0004$ | $1.166 \pm 0.0303$ | $\underline{0.958} \pm 0.0004$ | $2.135 \pm 0.0450$ | $\underline{0.934} \pm 0.0004$ | $2.793 \pm 0.0865$ |
| | $(0,0,50)$ | $0.987 \pm 0.0004$ | $0.899 \pm 0.0007$ | $\underline{1.785} \pm 0.0669$ | $0.911 \pm 0.0006$ | $\underline{3.814} \pm 0.0768$ | $0.793 \pm 0.0008$ | $\underline{3.046} \pm 0.2845$ |
| | $(10^8,0.1,50)$ | $\underline{0.991} \pm 0.0005$ | $\mathbf{0.973} \pm 0.0009$ | $\mathbf{1.813} \pm 0.0268$ | $\mathbf{0.969} \pm 0.0005$ | $\mathbf{4.000} \pm 0.0094$ | $\mathbf{0.945} \pm 0.0006$ | $\mathbf{3.662} \pm 0.1005$ |
| General-ODIN | $(0,0,0)$ | $0.988 \pm 0.0004$ | $0.769 \pm 0.0004$ | $0.506 \pm 0.0165$ | $0.719 \pm 0.0014$ | $0.816 \pm 0.0192$ | $0.605 \pm 0.0013$ | $0.558 \pm 0.1683$ |
| | $(10^6,0.1,0)$ | $\mathbf{0.995} \pm 0.0004$ | $\underline{0.951} \pm 0.0006$ | $1.461 \pm 0.0321$ | $\underline{0.960} \pm 0.0005$ | $3.007 \pm 0.0316$ | $\underline{0.940} \pm 0.0008$ | $2.619 \pm 0.1077$ |
| | $(0,0,50)$ | $0.981 \pm 0.0004$ | $0.859 \pm 0.0007$ | $\underline{1.814} \pm 0.0685$ | $0.803 \pm 0.0006$ | $4.204 \pm 0.0159$ | $0.826 \pm 0.0008$ | $\mathbf{4.014} \pm 0.2246$ |
| | $(10^6,0.1,50)$ | $0.990 \pm 0.0005$ | $\mathbf{0.971} \pm 0.0010$ | $\mathbf{1.835} \pm 0.0669$ | $\mathbf{0.963} \pm 0.0004$ | $\mathbf{4.287} \pm 0.0284$ | $\mathbf{0.951} \pm 0.0005$ | $3.695 \pm 0.1921$ |
| Energy | $(0,0,0)$ | $0.977 \pm 0.0006$ | $0.671 \pm 0.0012$ | $0.453 \pm 0.0121$ | $0.682 \pm 0.0012$ | $0.675 \pm 0.0148$ | $0.557 \pm 0.0014$ | $0.521 \pm 0.0131$ |
| | $(1,0.1,0)$ | $\mathbf{0.993} \pm 0.0005$ | $\mathbf{0.965} \pm 0.0004$ | $1.266 \pm 0.0319$ | $\mathbf{0.963} \pm 0.0004$ | $2.125 \pm 0.0413$ | $\mathbf{0.960} \pm 0.0003$ | $2.648 \pm 0.0596$ |
| | $(0,0,50)$ | $\underline{0.987} \pm 0.0005$ | $0.779 \pm 0.0010$ | $\mathbf{1.920} \pm 0.0725$ | $0.793 \pm 0.0009$ | $\mathbf{3.659} \pm 0.0659$ | $0.767 \pm 0.0010$ | $\mathbf{4.397} \pm 0.2165$ |
| | $(1,0.1,50)$ | $0.980 \pm 0.0005$ | $\underline{0.943} \pm 0.0005$ | $\underline{1.839} \pm 0.0662$ | $\underline{0.941} \pm 0.0005$ | $\underline{3.421} \pm 0.0619$ | $\underline{0.936} \pm 0.0005$ | $\underline{3.917} \pm 0.1691$ |
| Mahala-nobis | $(0,0,0)$ | $0.990 \pm 0.0007$ | $0.715 \pm 0.0011$ | $0.571 \pm 0.0110$ | $0.736 \pm 0.0011$ | $0.822 \pm 0.0165$ | $0.591 \pm 0.0011$ | $0.564 \pm 0.0203$ |
| | $(0.1,0.1,0)$ | $\mathbf{0.994} \pm 0.0004$ | $\underline{0.950} \pm 0.0009$ | $1.532 \pm 0.0351$ | $\underline{0.960} \pm 0.0010$ | $2.276 \pm 0.0466$ | $\underline{0.938} \pm 0.0004$ | $2.915 \pm 0.1132$ |
| | $(0,0,50)$ | $0.985 \pm 0.0004$ | $0.880 \pm 0.0005$ | $\underline{2.550} \pm 0.0681$ | $0.883 \pm 0.0006$ | $\underline{4.091} \pm 0.1013$ | $0.774 \pm 0.0007$ | $\underline{4.274} \pm 0.2305$ |
| | $(0.1,0.1,50)$ | $\underline{0.992} \pm 0.0006$ | $\mathbf{0.961} \pm 0.0005$ | $\mathbf{2.616} \pm 0.0857$ | $\mathbf{0.966} \pm 0.0005$ | $\mathbf{4.325} \pm 0.0055$ | $\mathbf{0.949} \pm 0.0003$ | $\mathbf{4.308} \pm 0.2011$ |

Table 1: **Results of concept learning with different parameter settings across various OOD detectors and test OOD datasets.** The ID dataset is AwA for both training and test, and the auxiliary OOD dataset is MSCOCO. Hyperparameters are in the order of $(\lambda_{\text{mse}}, \lambda_{\text{norm}}, \lambda_{\text{sep}})$, and their values are set based on the scale of corresponding regularization terms in Eqn. (7), for a specific choice of the OOD detector. Further investigation, including an ablation study on each regularization term can be found in Appendix B.2. Across the rows (for a given OOD detector and OOD dataset), the best value is **boldfaced**, and second best value is underscored. The 95% confidence intervals are estimated by bootstrapping the test set over 200 trials.

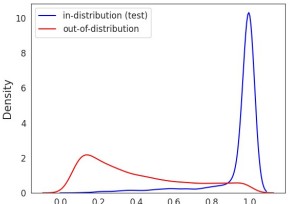 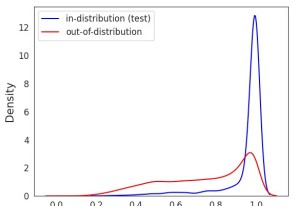 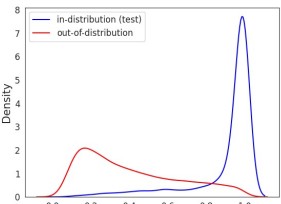

(a) Target distribution of $S(\mathbf{x}, \mathbf{f})$ in the canonical world.

(b) Reconstructed distribution of $S^{\mathrm{con}}(\mathbf{x}, \mathbf{f})$ in the concept world, using concepts by (Yeh et al., 2020).

(c) Reconstructed distribution of $S^{\mathrm{con}}(\mathbf{x}, \mathbf{f})$ in the concept world, using concepts by ours.

Figure 3: Detection completeness and estimated density of OOD score $S(\mathbf{x}, \mathbf{f})$ from MSP detector. Concepts by ours are learned using $\lambda_{\mathrm{mse}} = 10, \lambda_{\mathrm{norm}} = 0.1, \lambda_{\mathrm{sep}} = 50$. Comparison is made between AwA test set (ID; blue) vs. SUN (OOD; red).

## 4.1 EFFECTIVENESS OF OUR METHOD

In this subsection, we carry out experiments to answer the following question: *does our concept learning objective effectively encourage concepts to have the desired properties of detection completeness and concept separability?* Table 1 summarizes the results of concept learning with various combinations of hyperparameters for the proposed regularization terms in Eqn. (7): **i)** all the hyperparameters are set to $0$ (first row); **ii)** only the terms directly relevant to detection completeness (*i.e.*, reconstruction error $J_{\mathrm{norm}}(\mathbf{C}, \mathbf{g})$ and mean-squared error $J_{\mathrm{mse}}(\mathbf{C}, \mathbf{g})$) are included (second row); **iii)** only the term responsible for concept separability $J_{\mathrm{sep}}(\mathbf{C})$ is included (third row); **iv)** all the regularization terms are included (fourth row).

In all cases, we observe that the baseline achieves a high enough classification completeness score, but the lowest detection completeness and concept separability. This indicates that the *concepts discovered by (Yeh et al., 2020) would be sufficient to describe the DNN classifier, but using the same set of concepts would be inappropriate for the OOD detector*. In contrast, our method significantly improves the detection completeness (and even the classification completeness) by having $\lambda_{\mathrm{mse}} > 0, \lambda_{\mathrm{norm}} > 0$ and concept separability by having $\lambda_{\mathrm{sep}} > 0$, compared to the baseline. Moreover, the three terms make the best synergy together in almost all cases.

**Detection completeness and accurate reconstruction of $\mathcal{Z}$.** Additionally, we observe whether the proposed evaluation metrics are well-aligned with the interpretability of the resulting concept-based explanations. In Fig. 3, we observe that concepts by (Yeh et al., 2020) with low detection completeness ($\eta_{\mathbf{f}, S}(\mathbf{C}) = 0.782$ for MSP and $\eta_{\mathbf{f}, S}(\mathbf{C}) = 0.682$ for Energy) lead to a strong mismatch between the score distributions on both ID data and OOD data. In contrast, concepts learned by our method with high detection completeness ($\eta_{\mathbf{f}, S}(\mathbf{C}) = 0.961$ for the MSP detector, and $\eta_{\mathbf{f}, S}(\mathbf{C}) = 0.941$ for the Energy detector) approximate the target score distributions more closely on both ID data and OOD data. By reducing the performance gap of OOD detector between canonical world and concept world, it leads to more *accurate* explanations for OOD detectors.

## 4.2 CONCEPT-BASED EXPLANATIONS FOR OOD DETECTORS

**Contribution of each concept to detection.** The proposed concept learning algorithm learns concepts for both the classifier and OOD detector considering all the classes, and we address the following question: *how much does each concept contribute to the detection results for inputs predicted to a particular class?*. Recent works have adopted the Shapley value from Coalitional Game theory literature (Shapley, 1953; Fujimoto et al., 2006) for scoring the importance of a feature subset towards the predictions of a model (Chen et al., 2018; Lundberg & Lee, 2017; Sundararajan & Najmi, 2020). Extending this idea, we modify the characteristic function of the Shapley value using our per-class detection completeness metric (Eqn. (11) in Appendix A.2). The modified Shapley value of a concept $\mathbf{c}_i \in \mathbf{C}$ with respect to the predicted class $j \in [L]$ is defined as

$$\mathrm{SHAP}(\eta_{\mathbf{f}, S}^j, \mathbf{c}_i) := \sum_{\mathbf{C}' \subseteq \mathbf{C} \setminus \{\mathbf{c}_i\}} \frac{(m - |\mathbf{C}'| - 1)! \, |\mathbf{C}'|!}{m!} \left( \eta_{\mathbf{f}, S}^j(\mathbf{C}' \cup \{\mathbf{c}_i\}) - \eta_{\mathbf{f}, S}^j(\mathbf{C}') \right), \quad (9)$$

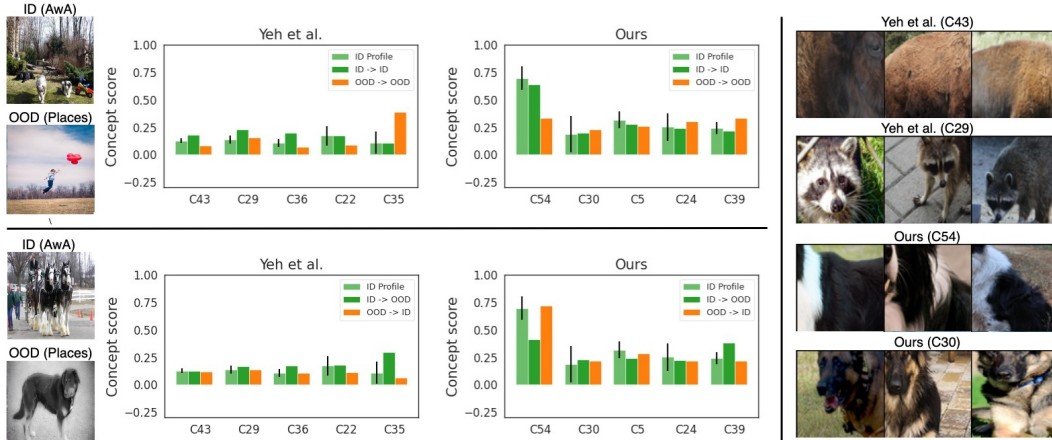

Figure 4: **Concept-based explanations for Energy OOD detector using concepts by Yeh et al. (2020) vs. ours**. Images are randomly selected from AwA test set (ID) and `Places` (OOD), and all predicted to class "Collie". ID profile shows the normal concept-score patterns for ID Collie images.

where $\mathbf{C}'$ is a subset of $\mathbf{C}$ excluding concept $\mathbf{c}_i$, and $\eta_{\mathbf{f},S}^j$ is the per-class detection completeness with respect to class $j$. This Shapley value captures the average marginal contribution of concept $\mathbf{c}_i$ towards explaining the decisions of the OOD detector for inputs that are predicted into class $j$.

Eventually, we interpret the behavior of the given OOD detector by plotting the concept score patterns with respect to the concepts ranked by the above Shapley importance score. Figure 4 illustrates the generated explanations given correctly-detected inputs (ID / OOD input detected as ID / OOD; first row of figure), and incorrectly-detected inputs (ID / OOD input detected as OOD / ID; second row of figure). Overall, we observe that the OOD detector predicts an input as ID when the concept scores show a similar pattern to the ID profile, or predicts an input as OOD when the concept score pattern is far from the ID profile. For instance, the fourth input is OOD image from `Places` dataset but detected as ID, since its score for C54 (furry dog skin) is as high as usual ID Collie images (which is true in the image). Thus, we conclude this to be an understandable mistake by the OOD detector.

We also provide qualitative comparison between Yeh et al. and our method in the resulting explanations for OOD detector. We observe that Yeh et al. fails to generate visually-distinguishable explanations between detected-ID and detected-OOD inputs. The separation between the solid green bars and the orange bars in each figure becomes more visible in our explanations, which enables more intuitive interpretation for human users, and this reflects our design goal for concept separability. It is also noteworthy that our concepts that are most important to distinguish ID Collie from OOD Collie (*i.e.*, C54 and C30) are more specific, and finer-grained characteristics of Collie, while Yeh et al. finds concepts that are vaguely similar to the features of dog, but rather generic (*i.e.*, C43 and C29). This is the reason we require more number of concepts to achieve detection completeness and concept separability, compared to solely considering the classification completeness [6].

## 5   CONCLUSION

In this work, we make a first attempt at developing an unsupervised and human-interpretable explanation method for black-box OOD detectors based on high-level concepts derived from the internal layer representations of a (paired) DNN classifier. We propose novel metrics viz. *detection completeness* and *concept separability* to evaluate the completeness (sufficiency) and quality of the learned concepts for OOD detection. The proposed concept learning method is quite general and applies to a broad class of off-the-shelf OOD detectors. Through extensive experiments and qualitative examples, we demonstrate the practical utility of our method for understanding and debugging an OOD detector. We discuss additional aspects of our method such as the choice of auxiliary OOD dataset, human subject study, and societal impact in Appendix E.

---

[6]In Figure 4, after concept learning with $m = 100$ and duplicate removal, we find 44 non-redundant concepts for Yeh et al. ($\lambda_{\mathrm{mse}} = \lambda_{\mathrm{norm}} = \lambda_{\mathrm{sep}} = 0$), and 100 distinct concepts for ours ($\lambda_{\mathrm{mse}} = 1$, $\lambda_{\mathrm{norm}} = 0.1$, $\lambda_{\mathrm{sep}} = 10$).

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

# Appendix

In Section A, we discuss the connection of the proposed concept separability to Bhattacharya Distance, and the per-class variations of detection completeness and concept separability, followed by the overall algorithm for concept learning. In Section B, we provide the detailed setup for the experiments. In Section C, we discuss whether our concept learning objective remains effective even when synthesized auxiliary OOD dataset similar to target ID data is used. In Section D, we illustrate additional examples of our concept-based explanations.

## A  CONCEPT LEARNING

### A.1  CONNECTION TO THE BHATTACHARYA DISTANCE

We note that the proposed separability metric in Section 3.2 is closely related to the Bhattacharya distance Bhattacharyya (1943) for the special case when the concept scores from both ID and OOD data follow a multivariate Gaussian density. The Bhattacharya distance is a well known measure of divergence between two probability distributions, and it has the nice property of being an upper bound to the Bayes error rate in the two-class case Fukunaga (1990a). For the special case when the concept scores from both ID and OOD data follow a multivariate Gaussian with a shared covariance matrix, it can be shown that the Bhattacharya distance reduces to the separability metric:

$$J_{\text{sep}}(\mathbf{C}) := J_{\text{sep}}(V_{\text{in}}(\mathbf{C}), V_{\text{out}}(\mathbf{C})) = \text{tr}\left[\mathbf{S}_w^{-1}\mathbf{S}_b\right]. \tag{10}$$

(ignoring scale factors).

### A.2  PER-CLASS DETECTION COMPLETENESS

**Per-Class Variations.** We propose per-class measures for the detection completeness, which are obtained by simply modifying $\eta_{\mathbf{f},S}(\mathbf{C})$ in Eqn. (2) and $J_{\text{sep}}(\mathbf{C})$ based on the subset of ID and OOD data whose predictions are class $y \in [L]$. We refer to these per-class variations as *per-class detection completeness* (denoted by $\eta_{\mathbf{f},S}^{y}(\mathbf{C})$).

**Definition 3.** Given a trained DNN classifier $\mathbf{f} = \mathbf{h} \circ \phi$, a trained OOD detector with score function $S(\mathbf{x}, \mathbf{f})$, and a set of concept vectors $\mathbf{C}$, the *detection completeness* relative to class $j \in [L]$ with respect to the ID distribution $P_{\text{in}}(\mathbf{x}, y)$ and OOD distribution $P_{\text{out}}(\mathbf{x})$ is defined as

$$\eta_{\mathbf{f},S}^{j}(\mathbf{C}) := \frac{\sup_{\mathbf{g}} \text{AUC}^{j}(\mathbf{h} \circ \widehat{\phi}_{\mathbf{g},\mathbf{C}}) - b_r}{\text{AUC}(\mathbf{h} \circ \phi) - b_r}, \tag{11}$$

where $\text{AUC}^{j}(\mathbf{h} \circ \widehat{\phi}_{\mathbf{g},\mathbf{C}})$ is the AUROC of the detector conditioned on the event that the class predicted by the concept-world classifier $\mathbf{h} \circ \widehat{\phi}_{\mathbf{g},\mathbf{C}}$ is $j$ (note that the denominator has the global AUROC). The baseline AUROC $b_r$ is equal to $0.5$ as before. This per-class detection completeness is used in the modified Shapley value defined in section 4.2.

### A.3  PER-CLASS CONCEPT SEPARABILITY

In section 3.2, we focused on the separability between the concept scores of ID and OOD data without considering the class prediction of the classifier. However, it would be more appropriate to impose a high separability between the concept scores on a per-class level. In other words, we would like the concept scores of detected-ID and detected-OOD data, that are predicted by the classifier into any given class $y \in [L]$ to be well separated. Consider the set of concept-score vectors from the detected-ID (or detected-OOD) dataset that are also predicted into class $y$:

$$V_{\text{in}}^{y}(\mathbf{C}) := \{\widetilde{\mathbf{v}}_{\mathbf{C}}(\mathbf{x}), \ \mathbf{x} \in D_{\text{in}}^{\text{tr}} \cup D_{\text{out}}^{\text{tr}} \ : \ \mathcal{D}_{\gamma}(\mathbf{x}, \mathbf{f}) = 1 \ \text{and} \ \widehat{y}(\mathbf{x}) = y\}$$

$$V_{\text{out}}^{y}(\mathbf{C}) := \{\widetilde{\mathbf{v}}_{\mathbf{C}}(\mathbf{x}), \ \mathbf{x} \in D_{\text{in}}^{\text{tr}} \cup D_{\text{out}}^{\text{tr}} \ : \ \mathcal{D}_{\gamma}(\mathbf{x}, \mathbf{f}) = 0 \ \text{and} \ \widehat{y}(\mathbf{x}) = y\}. \tag{12}$$

We can extend the definition of the global separability metric in Eq. (10) to a given predicted class $y \in [L]$ as follows

$$J_{\text{sep}}^{y}(\mathbf{C}) := J_{\text{sep}}(V_{\text{in}}^{y}(\mathbf{C}), V_{\text{out}}^{y}(\mathbf{C})) = \text{tr}\left[(\mathbf{S}_w^{y})^{-1}\mathbf{S}_b^{y}\right]$$

$$= (\boldsymbol{\mu}_{\text{out}}^{y} - \boldsymbol{\mu}_{\text{in}}^{y})^{T} (\mathbf{S}_w^{y})^{-1} (\boldsymbol{\mu}_{\text{out}}^{y} - \boldsymbol{\mu}_{\text{in}}^{y}). \tag{13}$$

We refer to these per-class variations as *per-class concept separability*. The scatter matrices $\mathbf{S}_w^y$ and $\mathbf{S}_b^y$ are defined similar to Eq. (3), using the per-class subset of concept scores $V_{\text{in}}^y(\mathbf{C})$ or $V_{\text{out}}^y(\mathbf{C})$, and the mean concept-score vectors from the detected-ID and detected-OOD dataset are also defined at a per-class level.

## A.4   Algorithm for Concept Learning

To provide the readers with a clear overview of the proposed concept learning approach, we include Algorithm 1. Note that in line 7 of Algorithm 1, the dimension reduction step in $V_{\text{in}}(\mathbf{C}) = \{\widetilde{\mathbf{v}}_{\mathbf{C}}(\mathbf{x}), \ \mathbf{x} \in D_{\text{in}}^{\text{tr}} \cup D_{\text{out}}^{\text{tr}} \ : \ \mathcal{D}_\gamma(\mathbf{x}, \mathbf{f}) = 1\}$ and $V_{\text{out}}(\mathbf{C}) = \{\widetilde{\mathbf{v}}_{\mathbf{C}}(\mathbf{x}), \ \mathbf{x} \in D_{\text{in}}^{\text{tr}} \cup D_{\text{out}}^{\text{tr}} \ : \ \mathcal{D}_\gamma(\mathbf{x}, \mathbf{f}) = 0\}$ involves the maximum function, which is not differentiable; specifically, the step $\widetilde{v}_{\mathbf{c}_i}(\mathbf{x}) = \max_{p,q} |\langle \phi^{p,q}(\mathbf{x}), \mathbf{c}_i \rangle|$. For calculating the gradients (backward pass), we use the log-sum-exp function with a temperature parameter to get a differentiable approximation of the maximum function, *i.e.*, $\max_{p,q} |\langle \phi^{p,q}(\mathbf{x}), \mathbf{c}_i \rangle| \approx \alpha \log \left[ \sum_{p,q} \exp \left( \frac{1}{\alpha} |\langle \phi^{p,q}(\mathbf{x}), \mathbf{c}_i \rangle| \right) \right]$ as $\alpha \to 0$. In our experiments, we set the temperature constant $\alpha = 0.001$ upon checking that the approximate value of $\widetilde{v}_{\mathbf{c}_i}(\mathbf{x})$ is sufficiently close to the original value using the maximum function.

---

**Algorithm 1** Learning concepts for OOD detector

---

**INPUT:** Entire training set $D^{\text{tr}} = \{D_{\text{in}}^{\text{tr}}, D_{\text{out}}^{\text{tr}}\}$, entire validation set $D^{\text{val}} = \{D_{\text{in}}^{\text{val}}, D_{\text{out}}^{\text{val}}\}$, classifier $\mathbf{f}$, detector $\mathcal{D}_\gamma$.
**INITIALIZE:** Concept vectors $\mathbf{C} = [\mathbf{c}_1 \cdots \mathbf{c}_m]$ and parameters of the network $\mathbf{g}$.
**OUTPUT:** $\mathbf{C}$ and $\mathbf{g}$.
  1: Calculate threshold $\gamma$ for $\mathcal{D}_\gamma$ using $D^{\text{val}}$ as the score at which true positive rate is $95\%$.
  2: **for** $t = 1, ...T$ epochs **do**
  3:   Compute the prediction accuracy of the concept-world classifier $\mathbf{f}^{\text{con}}$ using $D_{\text{in}}^{\text{tr}}$.
  4:   Compute the explainability regularization term as defined in Yeh et al. (2020).
  5:   Compute difference of feature representation between canonical world and concept world (i.e. $J_{\text{norm}}(\mathbf{C}, \mathbf{g})$).
  6:   Compute difference of detector outputs between canonical world and concept world using Eqn. (6).
  7:   Compute $V_{\text{in}}(\mathbf{C})$ and $V_{\text{out}}(\mathbf{C})$ using $D^{\text{tr}}, \mathcal{D}_\gamma$ and $\mathbf{C}$.
  8:   Compute separability between $V_{\text{in}}(\mathbf{C})$ and $V_{\text{out}}(\mathbf{C})$ using Eqn. (10) or Eqn. (13).
  9:   Perform a batch-SGD update of $\mathbf{C}$ and $g$ using Eqn. (7) as the objective.

---

## A.5   Accurate reconstruction of classifier outputs

We have performed additional experiments to understand if the proposed method can provide improvements in the classification setting. Let $\mathbf{C}_1$ denote the concept matrix learned by the method of Yeh et al. Let $\mathbf{C}_2$ denote the concept matrix learned by our method with $\lambda_{mse} = \lambda_{sep} = 0$ and $\lambda_{norm} = 0.1$ (set based on the scale of the regularization term $J_{norm}$). The idea is that we exclude the terms in the concept-learning objective (Eqn. 7) that depend on the OOD detector, but include the $\ell_2$ norm based reconstruction error of the layer representation. To evaluate the utility of these two sets of concepts for classification, we calculated the per-sample Hellinger distance between the predicted class probabilities of the original classifier and the concept-world classifier (based on either $\mathbf{C}_1$ or $\mathbf{C}_2$). Figure 5 compares the empirical distribution of the Hellinger distance for both sets of concepts $\mathbf{C}_1$ and $\mathbf{C}_2$. We observe that the distribution is more skewed towards zero with a higher density near zero and a shorter (right) tail in the case of $\mathbf{C}_2$ (red curve) compared to $\mathbf{C}_1$ (blue curve). This suggests that the class predictions are more accurately reconstructed by the concepts learned using our method with only the reconstruction error-based regularization. This can in-turn benefit the concept-based explanations for the classifier.

## A.6   Accurate reconstruction of OOD scores

In addition to Figure 3 where we compared the reconstruction accuracy of OOD scores using concepts by (Yeh et al., 2020) and ours, Figure 6 confirms that the same observation applies to Energy detector.

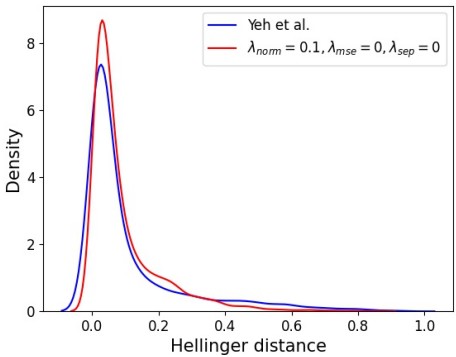

Figure 5: Examples for correct detection

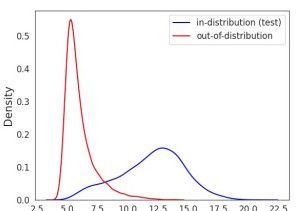 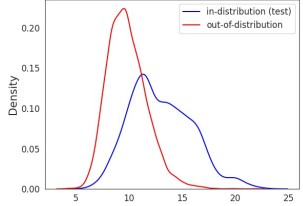 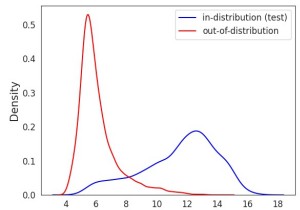

(a) Target distribution of $S(\mathbf{x}, \mathbf{f})$ in the canonical world.

(b) Distribution of $S^{\mathrm{con}}(\mathbf{x}, \mathbf{f})$ in the concept world, using concepts by (Yeh et al., 2020).

(c) Distribution of $S^{\mathrm{con}}(\mathbf{x}, \mathbf{f})$ in the concept world, using concepts by ours.

Figure 6: **Detection completeness and estimated density of OOD score $S(\mathbf{x}, \mathbf{f})$ from Energy detector.** Comparison is made between AwA test set (ID; blue) vs. SUN (OOD; red).

## A.7 CONCEPT SEPARABILITY AND VISUAL DISTINCTION IN EXPLANATIONS

In Fig. 7, we take the average of concept scores $V_{\mathrm{in}}(\mathbf{C})$ (or $V_{\mathrm{out}}(\mathbf{C})$) among the inputs that are predicted as class $y$, and detected as ID (or OOD) by Energy detector as an example. We can observe noticeably distinguishable pattern between detected-ID and detected-OOD concept scores when using concepts with higher concept separability ($J_{\mathrm{sep}}(\mathbf{C}, \mathbf{C}') = 3.421$), compared to those of low concept separability ($J_{\mathrm{sep}}(\mathbf{C}, \mathbf{C}') = 0.675$) by Yeh et al. (2020). These observations confirm our design motivation for the concept separability metric – that a higher value of the concept separability metric enables better *visual distinction* between the concept score patterns, suggesting better interpretability for humans.

## B IMPLEMENTATION DETAILS

We run all experiments with Tensorflow, Keras and NVIDIA GeForce RTX 2080Ti GPUs. We use test set bootstrapping with 200 runs to obtain the confidence interval for each hyperparameter set of concept learning.

### B.1 EXPERIMENTAL SETTING.

**OOD Datasets.** For the auxiliary OOD dataset for concept learning ($D_{\mathrm{out}}^{\mathrm{tr}}$), we use the unlabeled images from MSCOCO dataset (120K images in total) Lin et al. (2014). We carefully curate the dataset to make sure that no images contain overlapping animal objects with our ID dataset (*i.e.*, 50 animal classes of Animals-with-Attributes Xian et al. (2018)), then randomly sample 30K images. For OOD datasets for evaluation ($D_{\mathrm{out}}^{\mathrm{te}}$), we use the high-resolution image datasets processed by Huang and Li Huang & Li (2021).

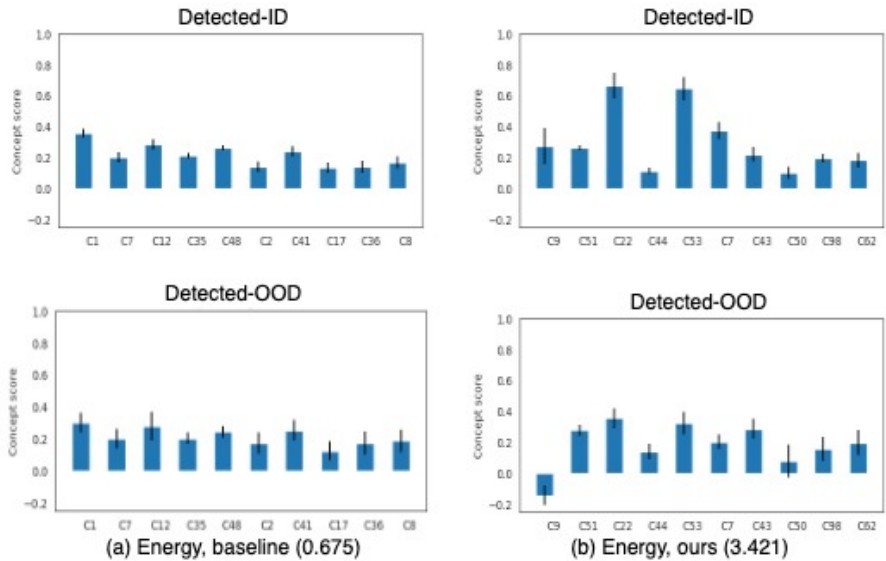

Figure 7: **Concept separability and visual distinction in the concept score patterns.** For the class "Giraffe", we compare the concept score patterns using two different sets of concepts. **Left:** Averaged scores of top-10 important concepts out of the concepts learned by Yeh et al. (2020)). **Right:** Averaged scores of top-10 important concepts out of the concepts learned by our method ($\lambda_{\mathrm{mse}} = 1$, $\lambda_{\mathrm{norm}} = 0.1$, $\lambda_{\mathrm{sep}} = 50$ with Energy detector). Concept importance is measured using the Shapley value of Eqn. (9).

**Hyperparameters for Concept Learning.** Throughout the experiments, we fix the number of concepts to $m = 100$ (unless specifically mentioned otherwise), and following the implementation of Yeh et al. (2020), we set $\lambda_{\mathrm{expl}} = 10$ and $\mathbf{g}$ to be a two-layer fully-connected neural network with 500 neurons in the hidden layer. We learn concepts based on feature representations from the layer right before the global max-pooling layer of the Inception-V3 model. After concept learning with $m$ concepts, we remove any duplicate (redundant) concept vectors by removing those with a dot product larger than $0.95$ with the remaining concept vectors Yeh et al. (2020).

## B.2 ADDITIONAL RESULTS

**Ablation study for concept learning.** We perform an ablation study that isolates the effect of each regularization term in our concept learning objective (Eqn. 7) towards our evaluation metrics: classification completeness, detection completeness, and relative concept separability. We also observe the coherency among the learned concepts by varying $\lambda_{\mathrm{mse}}$ and $\lambda_{\mathrm{sep}}$. Coherency of concepts was introduced by Ghorbani *et al.* Ghorbani et al. (2019) to ensure that the generated concept-based explanations are understandable to humans. It captures the idea that the examples for a concept should be similar to each other, while being different from the examples corresponding to other concepts. For the specific case of the image domain, the receptive fields most correlated to a concept $i$ (*e.g.*, "stripe pattern") should look different from the receptive fields for a different concept $j$ (*e.g.*, "wavy surface of sea"). Yeh *et al.* Yeh et al. (2020) proposed to quantify the coherency of concepts as

$$\frac{1}{m\,K} \sum_{i=1}^{m} \sum_{\mathbf{x}' \in T_{\mathbf{c}_i}} \langle \phi(\mathbf{x}'), \mathbf{c}_i \rangle, \tag{14}$$

where $T_{\mathbf{c}_i}$ is the set of $K$-nearest neighbor patches of the concept vector $\mathbf{c}_i$ from the ID training set $D_{\mathrm{in}}^{\mathrm{tr}}$.

We use this metric to quantify how understandable our concepts are for different hyperparameter choices. Figure 8 shows that aligned with our intuition, large $\lambda_{\mathrm{mse}}$ helps to improve the detection completeness. Having non-zero $\lambda_{\mathrm{mse}}$ is also helpful to improve the classification completeness even further, and surprisingly concept separability as well, without sacrificing the coherency of concepts. On the other hand, on the right side of Figure 8, we observe that large relative concept separability with large $\lambda_{\mathrm{sep}}$ comes at the expense of lower detection completeness and coherency. Recall that

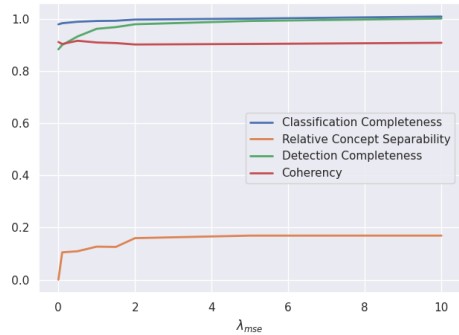

(a) Ablation study varying $\lambda_{\mathrm{mse}}$; we set $\lambda_{\mathrm{norm}} = 0.1, \lambda_{\mathrm{sep}} = 0$

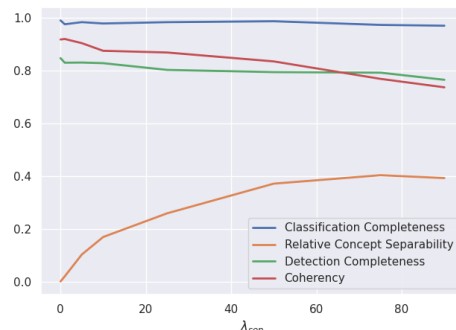

(b) Ablation study varying $\lambda_{\mathrm{sep}}$; we set $\lambda_{\mathrm{mse}} = 0, \lambda_{\mathrm{norm}} = 0$.

Figure 8: **Ablation study with respect to $J_{\mathbf{mse}}(\mathbf{C}, \mathbf{g})$ and $J_{\mathbf{sep}}(\mathbf{C})$.** We fix $m = 100, \lambda_{\mathrm{expl}} = 10$, and the OOD detector used for concept learing and evaluation is Energy Liu et al. (2020)

when visualizing what each concept represents for human's convenience, we apply threshold 0.8 to only presents (see Figure 10). Low coherency with respect to Eqn. 14 (*i.e.*, 0.768 with $\lambda_{\mathrm{sep}} = 75$) means that there are much less number of examples that can pass the threshold, meaning that users can hardly understand what the concepts at hand entails. This observation suggests that one needs to balance between concept coherency and concept separability depending on which property would be more useful for a specific application of concepts.

**Transferability of concepts across OOD detectors.** Our work essentially suggests to use different set of concepts for a specific target OOD detector, as $J_{\mathrm{mse}}(\mathbf{C}, \mathbf{g})$ and $J_{\mathrm{sep}}(\mathbf{C})$ in Eqn. (7) depend on a choice of OOD detector. In practice, however, one might not have enough computational capacity to prepare multiple sets of concepts for all type of OOD detectors at hand. Here, we inspect whether the concepts targeted for a certain type of OOD detector are also good to be used for other OOD detectors.

We explore the transferability of concepts targeted to MSP Hendrycks & Gimpel (2016) detector in Table 2a, and Energy Liu et al. (2020) in Table 2b. Not surprisingly, we observe that concepts targeted for Energy yields the best detection completeness score when tested with the same type of OOD detector, but still make meaningful improvement with other detectors as well. When it comes to relative concept separability, it is transferred even better across different OOD detectors. For instance, the concepts lead to $J_{\mathrm{sep}}(\mathbf{C}, \mathbf{C}') = 0.862$ with `Textures`, the best relative concept separability is achieved with ODIN detector (*i.e.*, $J_{\mathrm{sep}}(\mathbf{C}, \mathbf{C}') = 0.862$) and which is even higher than the best results we could obtain using the set of concepts targeted for ODIN (*i.e.*, $J_{\mathrm{sep}}(\mathbf{C}, \mathbf{C}') = 0.414$ with $\lambda_{\mathrm{mse}} = 0, \lambda_{\mathrm{norm}} = 0, \lambda_{\mathrm{sep}} = 50$ in Table 1).

| OOD dataset | Metrics | $\mathcal{D}$ | | | |
|---|---|---|---|---|---|
| | | MSP | ODIN | Energy | Mahal |
| Places | $\eta_{\mathbf{f},S}(\mathbf{C})$ | 0.959 | 0.952 | 0.938 | 0.947 |
| | $J_{\mathrm{sep}}(\mathbf{C}, \mathbf{C}')$ | 0.327 | 0.288 | 0.361 | 0.338 |
| SUN | $\eta_{\mathbf{f},S}(\mathbf{C})$ | 0.961 | 0.954 | 0.945 | 0.953 |
| | $J_{\mathrm{sep}}(\mathbf{C}, \mathbf{C}')$ | 0.266 | 0.294 | 0.390 | 0.351 |
| Textures | $\eta_{\mathbf{f},S}(\mathbf{C})$ | 0.938 | 0.946 | 0.932 | 0.930 |
| | $J_{\mathrm{sep}}(\mathbf{C}, \mathbf{C}')$ | 0.344 | 0.279 | 0.313 | 0.335 |
| iNaturalist | $\eta_{\mathbf{f},S}(\mathbf{C})$ | 0.946 | 0.946 | 0.933 | 0.930 |
| | $J_{\mathrm{sep}}(\mathbf{C}, \mathbf{C}')$ | 0.286 | 0.181 | 0.229 | 0.197 |

(a) Concepts targeted for MSP with $\lambda_{\mathrm{mse}} = 10, \lambda_{\mathrm{norm}} = 0.1, \lambda_{\mathrm{sep}} = 50$

| OOD data | Metrics | OOD detector | | | |
|---|---|---|---|---|---|
| | | MSP | ODIN | Energy | Mahal |
| Places | $\eta_{\mathbf{f},S}(\mathbf{C})$ | 0.956 | 0.954 | 0.971 | 0.954 |
| | $J_{\mathrm{sep}}(\mathbf{C}, \mathbf{C}')$ | 0.417 | 0.415 | 0.365 | 0.410 |
| SUN | $\eta_{\mathbf{f},S}(\mathbf{C})$ | 0.949 | 0.948 | 0.970 | 0.950 |
| | $J_{\mathrm{sep}}(\mathbf{C}, \mathbf{C}')$ | 0.355 | 0.286 | 0.400 | 0.353 |
| Textures | $\eta_{\mathbf{f},S}(\mathbf{C})$ | 0.931 | 0.943 | 0.964 | 0.947 |
| | $J_{\mathrm{sep}}(\mathbf{C}, \mathbf{C}')$ | 0.567 | 0.862 | 0.494 | 0.701 |
| iNaturalist | $\eta_{\mathbf{f},S}(\mathbf{C})$ | 0.943 | 0.939 | 0.973 | 0.940 |
| | $J_{\mathrm{sep}}(\mathbf{C}, \mathbf{C}')$ | 0.283 | 0.448 | 0.280 | 0.326 |

(b) Concepts targeted for Energy with $\lambda_{\mathrm{mse}} = 1, \lambda_{\mathrm{norm}} = 0.1, \lambda_{\mathrm{sep}} = 50$

Table 2: Transferability of concepts across different OOD detectors.

## C  DISCUSSION ON THE CHOICE OF AUXILIARY OOD DATASET IN CONCEPT LEARNING

Under circumstances where having access to auxiliary OOD dataset for concept learning is not feasible, we suggest that one could use generative methods to generate synthetic dataset, or apply data augmentation techniques (Hendrycks et al., 2022). Figure 9 shows an example of AwA image augmented by Hendrycks et al. (2022).

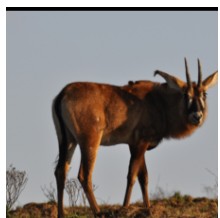 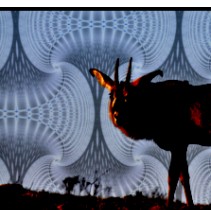

Figure 9: **Random example of augmented AwA dataset. Left:** original image in AwA train set. **Right:** corresponding image augmented by Hendrycks et al. (2022).

We evaluate the effectiveness of our concept learning objective when such augmented AwA train set is used as auxiliary OOD dataset. Table C illustrates that the generated concepts with augmented AwA (*i.e.*, OOD data close to target ID data) have comparable detection completeness and concept separability compared to when MSCOCO (*i.e.*, OOD data far from ID data) was used. But still, further evaluation on generated concept-based explanations with different choice of auxiliary OOD dataset remains as an interesting research question.

| OOD detector | Hyper-parameters | $\eta_{\mathbf{f}}(\mathbf{C}) \uparrow$ | Test OOD dataset | | | | | |
|---|---|---|---|---|---|---|---|---|
| | | | Places | | SUN | | Textures | |
| | | | $\eta_{\mathbf{f},S}(\mathbf{C}) \uparrow$ | $J_{\text{sep}}(\mathbf{C}, \mathbf{C}') \uparrow$ | $\eta_{\mathbf{f},S}(\mathbf{C}) \uparrow$ | $J_{\text{sep}}(\mathbf{C}, \mathbf{C}') \uparrow$ | $\eta_{\mathbf{f},S}(\mathbf{C}) \uparrow$ | $J_{\text{sep}}(\mathbf{C}, \mathbf{C}') \uparrow$ |
| Energy | $(1, 0.1, 50)$ | $0.955 \pm 0.0006$ | $0.940 \pm 0.0005$ | $1.746 \pm 0.0712$ | $0.9410 \pm 0.0005$ | $3.0703 \pm 0.0580$ | $0.927 \pm 0.0005$ | $3.417 \pm 0.1419$ |

Table 3: Results of concept learning with augmented AwA train set as auxiliary OOD in concept learning.

## D  EXPLANATIONS

### D.1  IMPORTANT CONCEPTS FOR EACH OOD DETECTOR

We show additional examples for the top-ranked concepts by $\text{SHAP}(\eta_{\mathbf{f},S}, \mathbf{c}_i)$ in Figure **??**. For each figure with a fixed choice of class prediction, we present receptive fields from ID test set corresponding to top concepts that contribute the most to the decisions of each OOD detector. All receptive fields passed the threshold test that the inner product between the feature representation and the corresponding concept vector is over $0.85$.

Moreover, in Fig. 11, we compare the important concepts discovered by the baseline method Yeh et al. (2020) (denoted "baseline") vs. ours. With the baseline, when the learned concepts are solely intended for reconstructing the behavior of the classifier, we observe that interpretation of both the classifier and OOD detector depends on a common set of concepts (*i.e.*, concepts 32, 10, and 47). On the other hand, the concepts learned by our method focus on reconstructing the behavior of both the OOD detector and the classifier. In this case, we observe that a distinct set of important concepts are selected for classification and OOD detection. We also observe that our method requires more concepts in order to address the decisions of both the classifier and OOD detector. For instance, the number of concepts obtained by our method and the baseline are 78 and 53 (respectively), out of a total 100 concepts after the duplicate removal of concept vectors. In short, when the concepts are only targeted at explaining the DNN classifier (as in the baseline Yeh et al. (2020)), the behavior of the OOD detector is merely described by the common set of concepts that are important for the DNN classifier. On the other hand, when not only the DNN classifier but also the OOD detector is taken into consideration during concept learning (*i.e.*, our method), we obtain a more diverse and

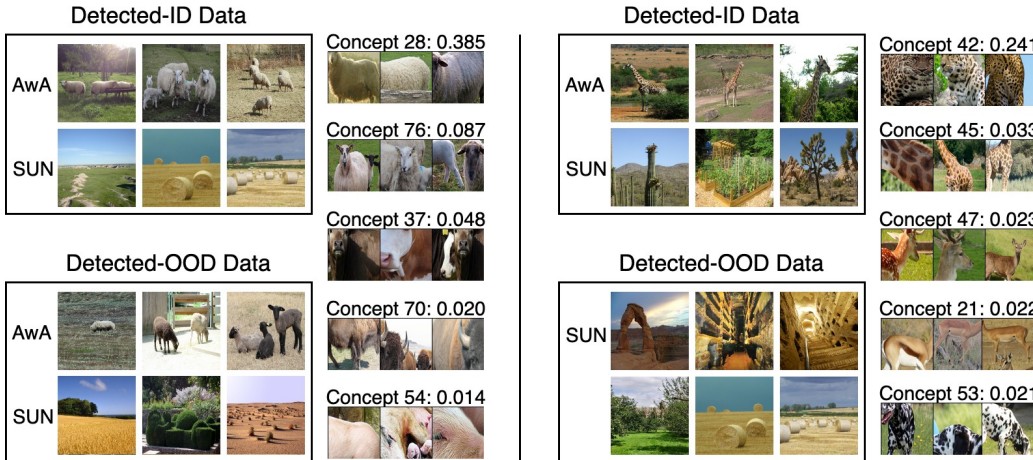

Figure 10: **Top-6 important concepts for Energy with respect. Left**: class "Sheep". **Right**: class "Giraffe"

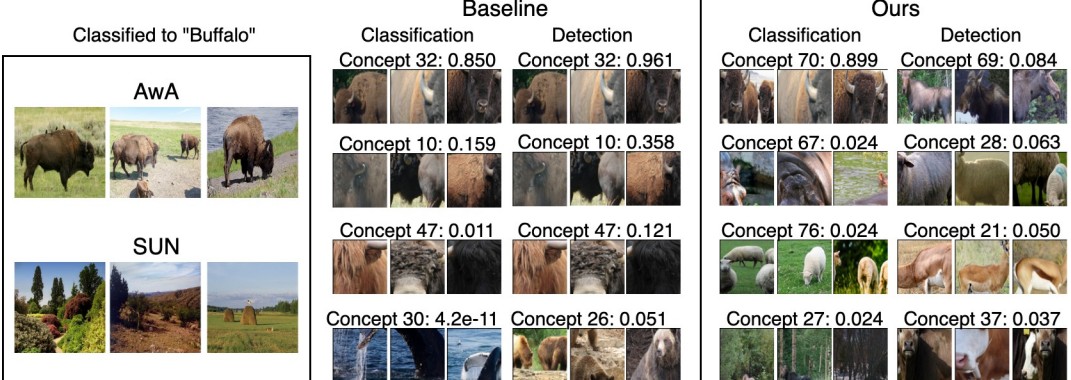

Figure 11: **Most important concepts for the Energy detector with respect to the predicted class "Buffalo".** We demonstrate randomly sampled images that are predicted by the classifier into this class. We compare the top-4 important concepts to describe the DNN classifier (and Energy detector), ranked by the Shapley value based on classification completeness $\text{SHAP}(\eta^j_{\mathbf{f}}, \mathbf{c}_i)$ (and detection completeness $\text{SHAP}(\eta^j_{\mathbf{f},S}, \mathbf{c}_i)$). "Baseline" corresponds to the case when the concepts are learned with $\lambda_{\text{mse}} = \lambda_{\text{norm}} = \lambda_{\text{sep}} = 0$, whereas "Ours" corresponds to the concepts learned with $\lambda_{\text{mse}} = 1, \lambda_{\text{norm}} = 0.1, \lambda_{\text{sep}} = 0$.

expanded set of concepts, and different concepts play a major role in interpreting the classification and detection results.

## D.2 MORE EXAMPLES OF OUR CONCEPT-BASED EXPLANATION

In Figure 12, we provide additional example of our concept-based explanation.

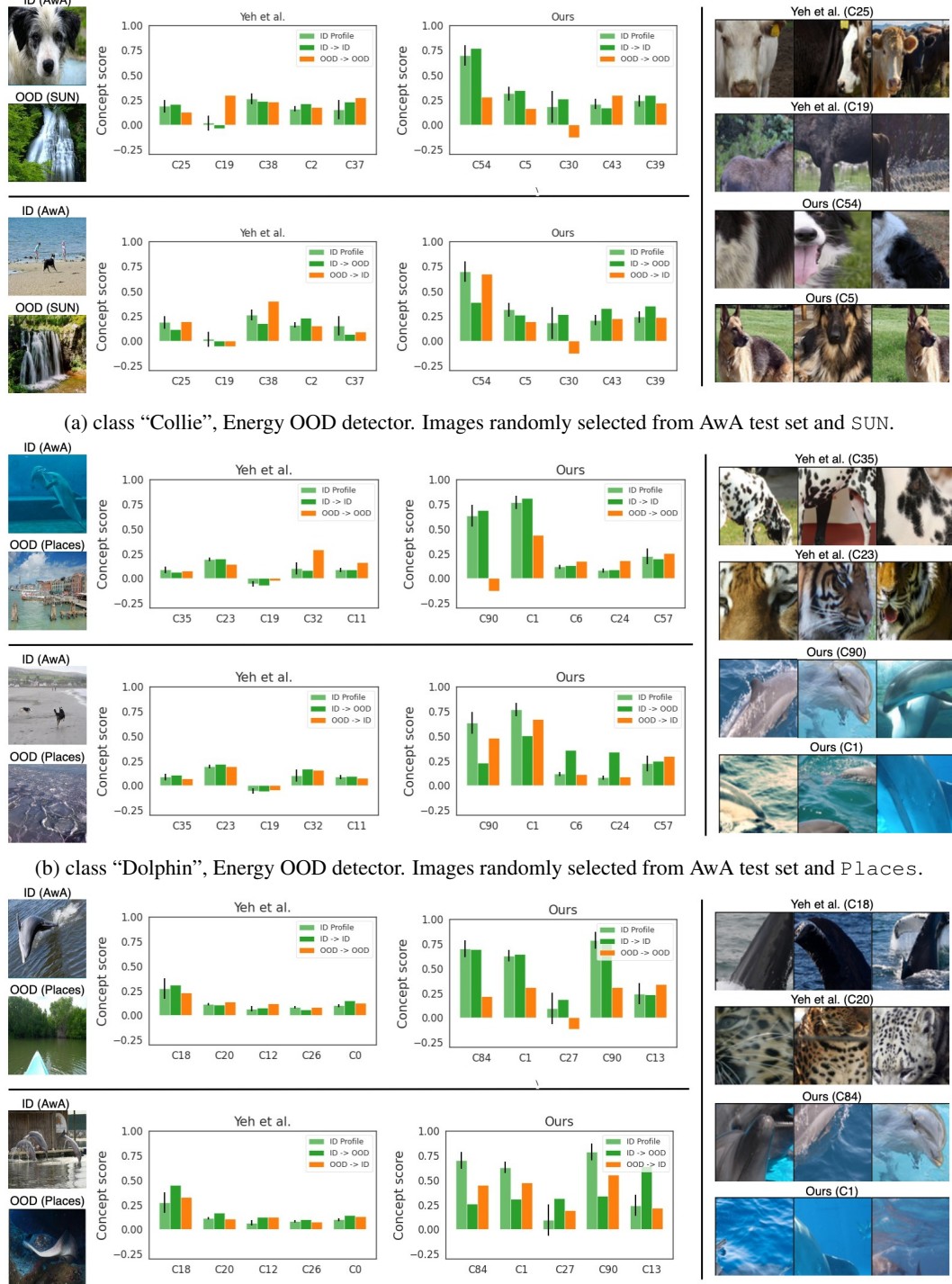

(a) class "Collie", Energy OOD detector. Images randomly selected from AwA test set and SUN.

(b) class "Dolphin", Energy OOD detector. Images randomly selected from AwA test set and Places.

(c) class "Dolphin", MSP OOD detector. Images randomly selected from AwA test set and Places.

Figure 12: **Concept-based explanations using concepts by Yeh et al. vs. ours.** ID profile shows the concept-score patterns for normal ID images.

### D.3 COUNTERFACTUAL ANALYSIS

To verify the important concepts identified by our modified Shapley value, we perform counterfactual analysis, addressing the following question: *if the OOD detector thought the input has different score for this concept, would the detection result be different?* As we do not assume to have groundtruth annotation for concepts, we construct concept score profiles of detected-ID (or detected-OOD) inputs from held-out ID (or OOD) dataset, and refer to this as ID (or OOD) concept profile. With the guidance of ID and OOD concept profiles, we take intervention on the concept scores of mis-detected inputs. Specifically, for ID data mis-detected as OOD, we update their concept scores using ID profiles, and similarlly, for OOD data mis-detected as ID,

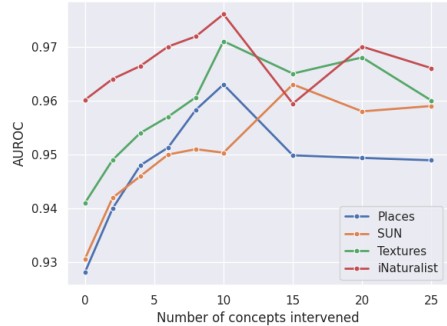

Figure 13: **Performance of MSP with test-time interventions on concept scoress.**

their concept scores are updated with OOD profiles. The number of concepts to be intervened can be varied. As shown in Figure 13, with intervention on more number of important concepts (ranked by $\text{SHAP}(\eta_{\mathbf{f},S}, \mathbf{c}_i))$), we observe an improved performance of OOD detector in concept world.

## E DISCUSSION AND SOCIETAL IMPACT

**Auxiliary OOD Dataset.** A limitation of our approach is its requirement of an auxiliary OOD dataset for concept learning, which could be hard to access in certain applications. To overcome that, a research direction would be to design generative models that simulate domain shifts or anomalous behavior and could create the auxiliary OOD dataset synthetically, allowing us additional control on the extent of distributional changes the resulting concepts could deal with (see Appendix C for further discussion).

**Human Subject Study.** Performing a human-subject (or user) study would be the ultimate way to evaluate the effectiveness of explanations, but remains largely unexplored even for in-distribution classification tasks. We emphasize that designing such a usability test with OOD detectors would be even more challenging due to the characteristics of the OOD detection task, compared to in-indistribution classification tasks. For in-distribution classifiers, users could potentially generate hypotheses about what high-level concepts should attribute to the class prediction, and compare their hypotheses to the provided explanations to determine the classifier's reliability. On the other hand, assessing the reliability of OOD detection involves checking whether a given input belongs to any of the natural distributions of concepts; this is essentially limited to whether users' mental models on such global distributions can be accurately probed via a couple of presented local instances. We believe that designing a thorough probing method for human interpretability on OOD detection would be an interesting yet challenging research quest by itself and our paper does not address that.

**Societal Impact.** Our work helps address the detection results of OOD detectors, giving practitioners the ability to explain the model's decision to invested parties. Our explanations can also be used to keep a data point as an understood mistake by the model rather than throwing it away without further analysis, which could help guide how to improve the OOD detector with respect to the concepts. However, this would also mean that more trust is put back into the human practitioner to not abuse the explanations or misrepresent them.

