# OpenReview forum: "Concept-based Explanations for Out-of-Distribution Detectors"
_ICLR.cc/2023/Conference — Submitted to ICLR 2023_

### Official Review · Reviewer_H68v · 2022-10-23

**Confidence:** 5
**Correctness:** 4
**Technical Novelty And Significance:** 3
**Empirical Novelty And Significance:** 2
**Recommendation:** 6

**Clarity, Quality, Novelty And Reproducibility:**

The paper builds heavily on Yeh et al. (2020) which is also related to concept-based explanation of neural networks with emphasis on the completeness desideratum, which is one of the two proposed metrics in this paper.  Specifically, the only difference between the original classification completeness of Yeh et al. and the paper is to replace the classification accuracy with an AUC score. The formulations on the concept space as well as the SHAP related evaluation also follow closely Yeh et al.

One key difference from Yeh et al. is the definition of concept separability score. The authors augment the regularization term from Yeh et al. with two other regularization terms $J_{norm}$ and $J_{mse}$, along with the LDA objective to encourage separability. Such a design makes sense to me, though compared to the original Yeh et al., the technical contribution tends to be limited.

While this paper focuses on OOD detection, I wonder if it would could also improve results in the simple classification case. The authors claim that Yeh et al. has potential issues in terms of reconstructing the features and may lead to degradation in performance. It would therefore be interesting to see if this is true.

The SHAP-based evaluation seems to also follow directly Yeh et al. While the novelty is limited, it is nice to see that the proposed method works well in combination with SHAP.

The definition of patch and receptive field is a bit vague. Why do you call it a $a_l \times b_l$ patch? Do you feed patches into a network rather than the whole image?


**Strength And Weaknesses:**

Strength

+ The paper is very well written and well organized. It is a good read.

+ Providing concept-based explanations to OOD detector is an interesting topic.

Weaknesses

- The notion of detection completeness is not entirely new. As shown in Definition 1 and 2, it builds on the classification completeness in Yeh et al. (2020).

- The proposed method has a limited scope in that it only focuses on explaining OOD detector.

- There are no direct comparisons between the proposed method and Yeh et al., which this paper heavily builds upon.


**Summary Of The Paper:**

This paper tackles the problem of concept-based explanations for deep neural network OOD detectors. The authors build on Yeh et al. (2020)’s argument on concept completeness and propose to use two metrics: detection (concept) completeness and concept separability. Following such metrics, the authors then design an algorithm for learning such desired concept-based explanations.

**Summary Of The Review:**

This paper tackles the problem of concept-based explanations for deep neural network OOD detectors. The authors build heavily on Yeh et al. (2020), with the key differences that (1) they introduce an additional metric, i.e., concept separability, (2) instead of focusing on classification, they focus on OOD detection. Other aspects of the paper including the completeness scores and SHAP-based evaluation are very similar to Yeh et al. Given the limited technical merit, I would place this paper marginally below the threshold.

---

> ### Comment · Reviewer_H68v · 2022-12-10
> **Thank you for the response**
>
> I have read the author response and other reviews. I would like to thank the authors for their response, which is helpful. Particularly, the added results and clarification did help strengthen the paper.
>
> The authors mentioned that the proposed method can be easily adapted to handle typical classification settings, which makes sense of course. However, this seems to also reflect the underlying similarity between the proposed method and Yeh et al.; in this regard, I somehow agree with Reviewer YcEH on the fair comparison between the proposed method and Yeh et al. Nevertheless, extending the concept-based method from Yeh et al. to OOD detection does count as a valid contribution.
>
> For the lack of user study mentioned by other reviewers, I feel this is less of an issue given that the main claim of contribution is on the concept-based explanations and that the paper follows closely Yeh et al.
>
> I am mostly satisfied with the response, and therefore would like to raise my score to 6.

---

> > ### Author Response · Authors · 2022-12-11
> > **Thank you**
> >
> > Thank you for increasing your score after carefully considering our responses and the other reviews. We are glad that the additional results and clarifications have helped improve the paper.

---

### Official Review · Reviewer_v8DY · 2022-10-25

**Confidence:** 4
**Correctness:** 3
**Technical Novelty And Significance:** 3
**Empirical Novelty And Significance:** 4
**Recommendation:** 6

**Clarity, Quality, Novelty And Reproducibility:**

The paper is mostly well-organized and written. The concept-based explanation also has its novelty on the OoD detection contexts, authors also make certain technical contributions to adapt concept-based explanations to the OoD contexts, which should be sound and feasible.

The evaluation also mostly supports the authors’ claims.

**Strength And Weaknesses:**

Strength:

- Important topic and work on OoD detection explanations
- Proposed techniques are general and could be applicable to various contexts, without the whitebox requirement
- Feasible solution with promising results

Weakness:

- Unclear about the robustness of the learned “concepts”
- Not very clear how to design the concept space or concept learning architecture
- Although High level and simple concept explanations are possible, it is not clear how such explanations could be helpful to guide the OoD detection developers in an actionable and constructive way.

**Summary Of The Paper:**

This paper proposes concept-based learning and an explanation for OoD Detection. The authors propose metrics to capture detection completeness and separability, which are used for the learning process. The evaluation of multiple datasets demonstrates the potential usefulness of the proposed techniques for OoD explanation.

**Summary Of The Review:**

Overall, I like the idea of this paper and think it works on an important problem, which does not receive enough attention previously. This paper makes an early step in explaining OoD detectors in relatively general settings without the requirement of white box. However, I still have a few concerns posted in the weakness part, that hope the authors could address during the rebuttal phase and add relevant discussion in the next version.

---

### Official Review · Reviewer_qdMm · 2022-10-27

**Confidence:** 4
**Correctness:** 3
**Technical Novelty And Significance:** 3
**Empirical Novelty And Significance:** 2
**Recommendation:** 5

**Clarity, Quality, Novelty And Reproducibility:**

Clarity: The paper could be more clear as to why concepts for OOD detectors are sensible.
Novelty: The idea depends heavily on existing concept-based explanation work, carefully applied to the OOD setting.
Quality: The evaluation is adequate.
Reproducibility: Adequate.

**Strength And Weaknesses:**

Strengths
- To my knowledge, this is the first work to study the notion of concept-based explanations for OOD detection.
- The idea of LDA on the concept layer is clever and simple.
- The proposed metrics are easy to understand. Detection completeness and concept separability naturally follow.

Weaknesses
- I would have expected more examples of what a concept-based explanation for an OOD detector means. Figure 1 is unconvincing. Please provide more qualitative evidence of the utility of concept-based explanations in the OOD setting.
- While the authors acknowledge the lack of a human subject experiment, it is not clear from the prose alone if concept-based explanations are even necessary (let alone helpful) for OOD detectors. I, among others, am unsure why concepts for OOD detectors. How does it help? A carefully designed user study would right this. It would also warrant a method that would lead to higher adoption and impact down the line.

**Summary Of The Paper:**

This paper proposes a method to interpret out-of-distribution detection using learnt high-level concepts.

**Summary Of The Review:**

The paper poses an interesting question of how to use concepts to generate explanations from OOD detectors. While the evaluation and idea is adequate, it is hard for me to recommend accept without a convincing user study or extensive qualitative examples of why such an explanation makes sense.

---

### Official Review · Reviewer_YcEH · 2022-10-28

**Confidence:** 4
**Correctness:** 3
**Technical Novelty And Significance:** 4
**Empirical Novelty And Significance:** 2
**Recommendation:** 6

**Clarity, Quality, Novelty And Reproducibility:**

The paper is clear, stating the addressed problem and the relation with existing works well. The conclusion section is not really reporting conclusions, but rather discussing limitations and societal impact of the work, leaving the reader pending final and concise answers to the posedresearch questions.

The quality of the paper and the research itself, including the novelty, is high, which inclines my evaluation towards the positive side. However, the experimental analysis and observations made with respect to existing works are unfair (see above). The authors should have focused on better highlighting the usefulness of the proposed concept-world and canonical-world representations, and the validation of the proposed metrics.

The work lacks reproducibility: I did not find mentions that the code will be available, neither details about the layers in between the proposed appraoch is deployed, and what effects this might have on the learned concepts and the overall explainability of the OOD detections. In its current form, the paper makes difficult to re-implement or even deploy the proposed approach to replicate the experiments.

**Strength And Weaknesses:**

__Strenghts__
- the problem addressed is relevant, and ground motivations for the research are discussed
- the paper is well written and carefully constructed, easy-to-follow and clear
- the new metrics are technically sound
- experiments in combination with existing methods

__Weaknesses__
- In my opinion, the main concern/weakness is in the way the authors analyse and interpret the results: the authors stress the fact that existingmethods are not able to guarantee concept separability and detection completeness, which are metrics that they define in this work. On the contrary, the method based on learning concepts by optimizing these metrics is performing better. There is no surprise in these results, as previous methods are not optimized for the considered metrics. Thus, it seems to me that the experimentations (in the way it is presented) is unfair.

- Another weakness is in the lack of details about the deployement of their approach within existing approaches: it seems a concept-based learning component (optimized with the proposed metrics) is deployed within intermediate layers, but no details about the 'where' in the network (which layer) it is used, and how it may change if deployed in different parts.

- Connected to above, if this can be deployed in different parts of the network, I imagine that different concepts are learned, of different semantics. How does it relate with the 'high-level' concepts that the authors aim at learning? Also, how the 'high-level' is defined?

**Summary Of The Paper:**

The paper addresses the problem of explaining the decisions of an out-of-distribution (OOD) detector by using concept-based representations. In order to learn concept-based explanation, the authors propose two measures, namely the concept detection completeness and the concept separability, which they optimize during training of OOD detectors.

**Summary Of The Review:**

The paper presents a novel idea, which is presented highlighting its relevance. The research is motivated and carefully constructed. The experimental analysis, although with several existing methods and on relevant datasets, brings to unfair statements in comparison with other methods, and lacks addressing strongly the validity of the proposed measures. Also, a more thorough analysis of the learned concepts should be provided: how are they relevant and what do they cover?
Overall is a good work, but I would like to see the response of the authors to my questions and doubts.

---

### Decision · Program_Chairs · 2023-01-20

**Decision:**

Reject

**Justification For Why Not Higher Score:**

The main issue is that the paper did not demonstrate the utility for concepts to guide the OoD detection developers in an actionable and constructive way, which could have been shown with a case study (can be a user study or some examples of how this explanation can help a real-world OOD detector).

**Justification For Why Not Lower Score:**

The paper tackles an under-looked topic, and extend the previous method well.

**Metareview: Summary, Strengths And Weaknesses:**

(a). The paper addresses the problem of explaining the decisions of an out-of-distribution (OOD) detector by using concept-based representations. In order to learn concept-based explanation, the authors propose two measures, namely the concept detection completeness and the concept separability, which they optimize during training of OOD detectors. They demonstrate that their method can better explain the OOD detectors compared to baseline methods by qualitative examples.
(b). The paper extended a previous work to explain OOD detector, where concept explanations were not previously explored. They show that they can produce better explanations qualitatively by considering the OOD detector jointly with the base model.
(c). Main issue: The utility of the proposed OOD concept explanations are not demonstrated. Detailed complaints from reviewers: The technical novelty is a bit weaker, and there is no user study for the paper to have a strong empirical result. The explanation of comparison with previous work is also a bit unclear, as they compare on many terms that are explicitly optimized in their paper (which makes the comparison unfair); they could have made this a bit more clear in the writings.

**Summary Of Ac-Reviewer Meeting:**

qdMm asked for a user study in initial review, but did not responded for the rest of the discussion period.
H68v was happy with the author's response and increased the score, and thought that a user study was not necessary.
YcEH was in general positive, but requested that the authors to highlight that what they do extra is not comparable to previous methods (Yeh et al.).
v8DY was unsatisfied that the author did not address questions regarding robustness of concepts and the utility for concepts to guide the OoD detection developers in an actionable and constructive way (and was leaning to lower the score).

My main decision is made by the fact that two reviewers (qdMm and v8DY) was unsatisfied about the utility of the OOD concept explanations. While conducting user studies or demonstrating the utility of explanations by as case-study may be time-consuming, it would greatly improve our confidence that the explanations proposed by the authors are actually helpful to developers of OOD detectors.